# Reward contingency gates selective cholinergic suppression of amygdala neurons

Eyal Y Kimchi[1,2]*, Anthony Burgos-Robles[1,3], Gillian A Matthews[1], Tatenda Chakoma[1], Makenzie Patarino[1], Javier C Weddington[1], Cody Siciliano[1,4], Wannan Yang[1], Shaun Foutch[1], Renee Simons[1], Ming-fai Fong[1,5], Miao Jing[6], Yulong Li[7], Daniel B Polley[8,9], Kay M Tye[1,10]*

[1]The Picower Institute for Learning and Memory, Department of Brain and Cognitive Sciences, Massachusetts Institute of Technology, Cambridge, United States; [2]Department of Neurology, Northwestern University, Chicago, United States; [3]The Department of Neuroscience, Developmental, and Regenerative Biology, Neuroscience Institute & Brain Health Consortium, University of Texas at San Antonio, San Antonio, United States; [4]Vanderbilt Center for Addiction Research, Department of Pharmacology, Vanderbilt University, Nashville, United States; [5]Coulter Department of Biomedical Engineering, Georgia Tech & Emory University, Atlanta, United States; [6]Chinese Institute for Brain Research, Beijing, China; [7]State Key Laboratory of Membrane Biology, Peking University School of Life Sciences; PKU-IDG/McGovern Institute for Brain Research; Peking-Tsinghua Center for Life Sciences, Beijing, China; [8]Eaton-Peabody Laboratories, Massachusetts Eye and Ear, Boston, United States; [9]Department of Otolaryngology – Head and Neck Surgery, Harvard Medical School, Boston, United States; [10]HHMI Investigator, Member of the Kavli Institute for Brain and Mind, and Wylie Vale Professor at the Salk Institute for Biological Studies, La Jolla, United States

*For correspondence:
eyal.kimchi@northwestern.edu (EYK);
tye@salk.edu (KMT)

**Competing interest:** The authors declare that no competing interests exist.

**Abstract** Basal forebrain cholinergic neurons modulate how organisms process and respond to environmental stimuli through impacts on arousal, attention, and memory. It is unknown, however, whether basal forebrain cholinergic neurons are directly involved in conditioned behavior, independent of secondary roles in the processing of external stimuli. Using fluorescent imaging, we found that cholinergic neurons are active during behavioral responding for a reward – even prior to reward delivery and in the absence of discrete stimuli. Photostimulation of basal forebrain cholinergic neurons, or their terminals in the basolateral amygdala (BLA), selectively promoted conditioned responding (licking), but not unconditioned behavior nor innate motor outputs. In vivo electrophysiological recordings during cholinergic photostimulation revealed reward-contingency-dependent suppression of BLA neural activity, but not prefrontal cortex. Finally, ex vivo experiments demonstrated that photostimulation of cholinergic terminals suppressed BLA projection neuron activity via monosynaptic muscarinic receptor signaling, while also facilitating firing in BLA GABAergic interneurons. Taken together, we show that the neural and behavioral effects of basal forebrain cholinergic activation are modulated by reward contingency in a target-specific manner.

## eLife assessment

This **valuable** article examines the role of basal forebrain cholinergic (ACh) projection neurons and their inputs to the basolateral amygdala (BLA) and effects on BLA activity during reward seeking.

The article provides **compelling** evidence that ACh may have different effects on network activity in the BLA depending on the state of the network during reward engagement, whereas behavioral data indicating that these ACh neurons/inputs are involved in uncued reward seeking specifically is somewhat less complete. The article will be of interest to those studying amygdala circuitry, reward processing, and neuromodulation broadly defined.

## Introduction

Acetylcholine (ACh) is a powerful neuromodulator thought to influence how the brain processes and learns about external stimuli (*Ballinger et al., 2016*; *Higley and Picciotto, 2014*; *Likhtik and Johansen, 2019*; *Newman et al., 2012*). The basal forebrain is a prominent source of cholinergic innervation of the entire cortical mantle, as well as related telencephalic structures such as the amygdala (*Gielow and Zaborszky, 2017*; *Li et al., 2018*). Most work on basal forebrain cholinergic circuits to date has focused on how ACh modifies the processing of other stimuli, either by increasing attention to conditioned stimuli (*Bakin and Weinberger, 1996*; *Baxter and Chiba, 1999*; *Gritton et al., 2016*; *Pinto et al., 2013*) or by enhancing associations between conditioned stimuli and unconditioned stimuli/reinforcers in learning and memory (*Ballinger et al., 2016*; *Crouse et al., 2020*; *Guo et al., 2019*; *Jiang et al., 2016*; *Letzkus et al., 2011*; *Sturgill et al., 2020*). In contrast, less work has been done to explore the behavioral effects of basal forebrain cholinergic neuron activation in the absence of external stimuli (*Aitta-Aho et al., 2018*). As a result, in most models of cholinergic function, the behavioral relevance of basal forebrain cholinergic neuron activity in the absence of discrete external stimuli is comparatively less specified (*Hasselmo and Sarter, 2011*; *Thiele and Bellgrove, 2018*).

Basal forebrain cholinergic projections have considerable physiological effects on postsynaptic neurons ex vivo, where sensory stimuli cannot be presented (*Kalmbach et al., 2012*; *Venkatesan et al., 2020*). Basal forebrain cholinergic projections have prominent direct effects on limbic circuits (*Venkatesan et al., 2020*), including the basolateral amygdala (BLA) (*McDonald and Mascagni, 2011*; *McDonald and Mascagni, 2010*; *Unal et al., 2015*). Ex vivo optogenetic activation of the cholinergic projection to the BLA affects postsynaptic activity differentially, suppressing neural activity at low firing rates (*Unal et al., 2015*). In contrast, however, in vivo cholinergic activation may increase spontaneous BLA neural firing (*Jiang et al., 2016*), suggesting that the postsynaptic effects of basal forebrain cholinergic projections may be depend upon as yet unidentified influences. We therefore sought to understand how basal forebrain cholinergic neuron activation, in the absence of discrete, environmental stimuli, may affect both behavior and downstream neural activity. To do this, we developed a new behavioral paradigm allowing us to study the short-time scale effects of basal forebrain cholinergic neurons on behavior in the absence of other discrete external cues. We sought to identify features that may regulate the effects of acetylcholine on behavior and downstream neural circuit activity.

## Results

### Establishing conditioned behavior in the absence of discrete cues

To study whether basal forebrain cholinergic neurons affect conditioned behavior in the absence of discrete cues, we developed a new behavioral paradigm, which we refer to as the Windows of Opportunity Task (WoOT) (*Figure 1A*). During WoOT, mice were head-fixed mice in front of a spout, with a goal of establishing a steady level of behavioral engagement, as measured by licking. WoOT sessions were divided into 'trials' of 3 s *Windows of Opportunity*, with intervening variable intertrial intervals (ITIs). Rewards were only delivered if mice licked the spout during an unsignaled *Window of Opportunity*. Because rewards were only delivered *after* the first lick in a *Window of Opportunity*, and because windows were not discretely cued and occurred with variable ITIs, mice did not know whether any given lick would be rewarded. The underlying task structure of this task is an operant, variable-interval reinforcement schedule with limited hold (*Ferster and Skinner, 1957*). In this operant schedule, unsignaled reward opportunities become available after a variable interval (ITI), but only for a limited time ('hold' in original terminology, 'window' in our terminology here).

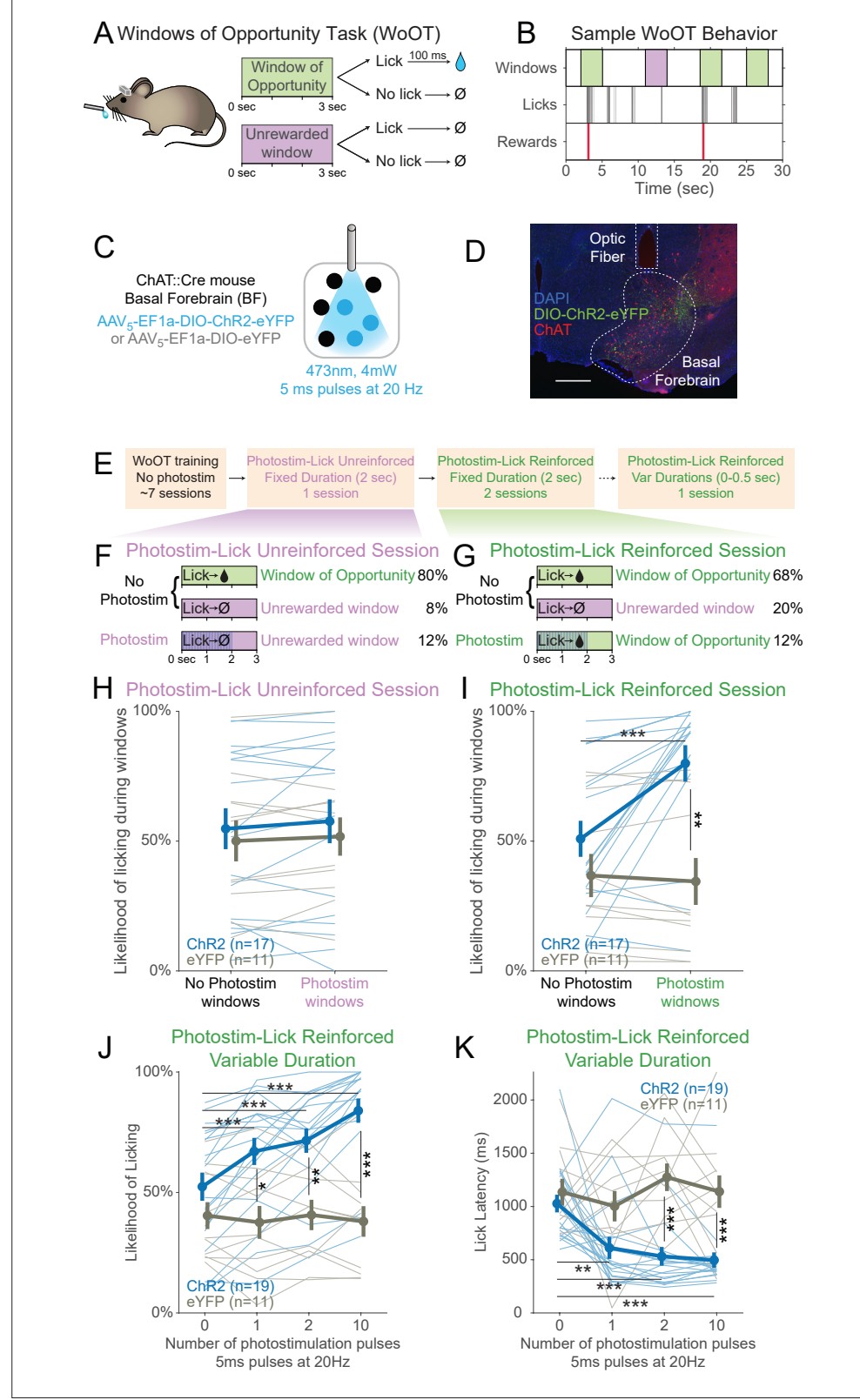

**Figure 1.** Photostimulation of basal forebrain cholinergic neurons promotes conditioned responding when associated with the opportunity to collect rewards. (**A**) Window of Opportunity Task (WoOT) to study conditioned responding in the absence of discrete cues. Mice were trained prior to any photostimulation, using an operant, variable-interval reinforcement schedule with limited hold (***Ferster and Skinner, 1957***). Sessions were divided

*Figure 1 continued*

into trials of 3 s *Windows of Opportunity*, with variable intertrial intervals (ITIs). Rewards were only delivered if mice licked during an unsignaled *Window of Opportunity* (green; 90% of trial windows). We also included a subset of *Unrewarded Windows*, on which, even if the mouse licked, reward would not be delivered (purple; 10% of trial windows), similar to intervening ITIs (white). Because windows were not discretely cued and occurred after variable ITIs, mice did not know when they initiated a lick whether it would be rewarded. (**B**) Sample WoOT behavior. If a mouse licked during an uncued *Windows of Opportunity* (green), then reward was delivered. But if a mouse licked during an uncued *Unrewarded Window* (purple) or during the ITI (white), reward was not delivered. reward delivery. The use of *Unrewarded Windows* (purple) allowed us to investigate behavior during epochs temporally matched to *Windows of Opportunity* (green). Training mice in this manner conditioned them to lick the spout with a relatively stable pattern of intermittent lick bouts (***Figure 1—figure supplement 1***). (**C**) Optogenetic strategy to photostimulate cholinergic basal forebrain neurons, by driving Cre-dependent expression of Channelrhodopsin-2 (ChR2) or a control fluorophore (eYFP) in mice expressing Cre-recombinase under control of the choline acetyltransferase promoter (ChAT::Cre). The photostimulation parameters displayed were used in later behavioral sessions. (**D**) Sample histology of fiber placement over cholinergic neurons in the posterior portion of the basal forebrain, the sublenticular substantia innominata/extended amygdala. Blue = DAPI nucleic acid staining, green = Cre-dependent expression of ChR2 fused to eYFP, red = anti-ChAT immunohistochemical staining. AP coordinate = –0.46. Scale bar = 500 microns. See also Figure S1A and B. (**E**) Behavioral training and testing sessions. After early WoOT training without any photostimulation, mice subsequently received testing during a Photostim-Unreinforced or Photostim-Reinforced sessions. (**F**) Photostim-Unreinforced session. In addition to no photostimulation *Windows of Opportunity* (green, 80%) and *Unrewarded Windows* (purple, 8%), mice received 2 s of photostimulation (blue lines) during a subset of *Unrewarded Windows* (purple, 12%) to study innate responses to photostimulation. (**G**) Photostim-Reinforced session. Conversely to Photostim-Unreinforced sessions, during Photostim-Reinforced sessions, photostimulation (blue lines) was delivered during a subset of *Windows of Opportunity* (green, 12%), during which, if mice licked, they would receive a reward. Mice still had many more *Windows of Opportunity* with no photostimulation (68%). (**H, I**) The likelihood of licking depended on an interaction of Virus, Photostimulation Window type, and Photostim-Reinforcement Session type (linear mixed effects model, $F_{1,78} = 5.26$, p=0.025). Thin lines represent data from all individual mice, pooled data are represented as mean ± SEM. During the Photostim-Unreinforced session (**H**), there was no significant difference between ChR2 (blue) and eYFP (gray) groups, regardless of photostimulation window type. However, during the Photostim-Reinforced session (**I**), ChR2 mice licked significantly more during Photostim windows than No Photostim windows (post hoc tests with Sidak correction for multiple comparisons: ***p<0.0001). ChR2 mice also licked significantly more than eYFP mice during Photostim windows during Photostim-Reinforced sessions (**p=0.001). There were no detectable effects of photostimulation on licking in eYFP mice, and no detectable differences in licking on no photostimulation windows between ChR2 and eYFP mice (all other post hoc comparisons p>0.10). (**J**) We initially photostimulated with a fixed duration, but in a follow-up session photostimulated with varied durations (0–0.5 s, using different number of pulses at the same frequency/duty cycle). The likelihood of licking depended on an interaction of Virus and the number of photostimulation pulses (linear mixed effects model, $F_{3,84} = 16.22$, p=2.1 × 10$^{-8}$). ChR2 mice licked significantly more on windows with 1, 2, or 10 pulses than windows without photostimulation, and more than eYFP mice with any number of pulses (post hoc tests with Sidak correction for multiple comparisons: *p=0.015, **p=0.0095, ***p<0.001; all other post hoc comparisons p>0.10). (**K**) When tested with different durations of photostimulation, the latency of the first lick depended on an interaction of Virus and the number of photostimulation pulses (linear mixed effects model, $F_{3,84} = 4.53$, p=0.005). ChR2 mice licked significantly sooner on windows with 1, 2, or 10 pulses than windows without photostimulation, and sooner than eYFP mice for 2 and 10 pulses (post hoc tests with Sidak correction for multiple comparisons: **p=0.0047, ***p<0.001; all other post hoc comparisons p>0.10).

The online version of this article includes the following source data and figure supplement(s) for figure 1:

**Source data 1.** The likelihood of licking during (or outside of) the photostim windows during the Photostim-Unreinforced session, as shown in ***Figure 1H***.

**Source data 2.** The likelihood of licking during (or outside of) the photstim windows during the Photostim-Reinforced session, as shown in ***Figure 1I***.

**Source data 3.** The likelihood of licking based on the number of photostimulation pulses during the Photostim-Reinforced Variable Duration, as shown in ***Figure 1J***.

**Source data 4.** The lick latency based on the number of photostimulation pulses during the Photostim-Reinforced Variable Duration, as shown in ***Figure 1K***.

**Figure supplement 1.** Training history was similar for optogenetic experimental ChR2 mice and control eYFP mice.

**Figure supplement 1—source data 1.** The number of training sessions in days prior to photo-stimulation, as

*Figure 1 continued on next page*

*Figure 1 continued*

shown in *Figure 1—figure supplement 1C*.

**Figure supplement 1—source data 2.** The likelihood of licking prior to photo-stimulation, as shown in *Figure 1—figure supplement 1D*.

**Figure supplement 1—source data 3.** The number of rewards earned in a session, prior to photo-stimulation, as shown in *Figure 1—figure supplement 1E*.

**Figure supplement 1—source data 4.** The session licking rate for each animal, prior to photo-stimulation, as shown in *Figure 1—figure supplement 1F*.

**Figure supplement 1—source data 5.** The percent lick per bout for each animal, prior to photo-stimulation, as shown in *Figure 1—figure supplement 1G*.

**Figure supplement 1—source data 6.** The median number of licks per bout for each animal, prior to photo-stimulation, as shown in *Figure 1—figure supplement 1H*.

**Figure supplement 1—source data 7.** The median inter bout interval in seconds for each animal, prior to photo-stimulation, as shown in *Figure 1—figure supplement 1I*.

**Figure supplement 2.** Responses across the Windows of Opportunity Task (WoOT) photostimulation testing sessions.

**Figure supplement 2—source data 1.** The likelihood of licking during the three sessions, as shown in *Figure 1—figure supplement 2A*.

**Figure supplement 2—source data 2.** The lick latencies during the three sessions, as shown in *Figure 1—figure supplement 2B*.

**Figure supplement 2—source data 3.** The likelihood of licking based on photostim on a previous trial during the different sessions, as shown in *Figure 1—figure supplement 2C*.

**Figure supplement 2—source data 4.** The likelihood of licking based on including an additional Photostim-Unreinforced session, as shown in *Figure 1—figure supplement 2D*.

**Figure supplement 2—source data 5.** The likelihood of licking based on including an additional Photostim-Unreinforced session after Photostim-Reinforced training, as shown in *Figure 1—figure supplement 2E*.

**Figure supplement 3.** Peri-event licking across the Windows of Opportunity Task (WoOT) photostimulation testing sessions.

**Figure supplement 4.** Photostimulation of basal forebrain cholinergic neurons increases arousal, but does not increase unconditioned movement and is not inherently reinforcing.

**Figure supplement 4—source data 1.** The percentage of time spent in in the laser ON side during the RTPP, as shown in *Figure 1—figure supplement 4F*.

**Figure supplement 4—source data 2.** The mean velocities and time spent in the center of the open field test before, during and after basal forebrain cholinergic photostimulation as shown in *Figure 1-figure supplement 4C, D*.

**Figure supplement 4—source data 3.** The mean velocities during Photostim-Unreinforced and Photostim-Reinforced sessions, as shown in *Figure 1-figure supplement 4H*.

**Figure supplement 4—source data 4.** The percent change of the pupil after photostim, as shown in *Figure 1-figure supplement 4J, L*.

Training mice in this manner conditioned them to lick the spout with a relatively stable pattern of intermittent lick bouts (*Figure 1B*), which we designate as 'conditioned responding,' using the formal operant sense (*Ferster and Skinner, 1957*). In addition to the *Windows of Opportunity*, We also defined a series of *Unrewarded Windows*, on which, even if the mouse licked, reward would not be delivered – equivalent to the intervening ITI (*Figure 1A and B*). While these *Unrewarded Windows* practically functioned as part of the ITI, they were programmatically necessary to study behavior in a way that matched the temporal characteristics of the *Windows of Opportunity*, but when reward was not available.

Prior to initial training, mice underwent surgery to prepare them for head-fixation and subsequent photostimulation of basal forebrain cholinergic neurons using Cre-dependent expression of either optogenetic Channelrhodopsin-2 (ChR2) or a control fluorophore (enhanced yellow fluorescent protein, eYFP) (*Figure 1C and D*). Fibers were implanted to target the sublenticular substantia innominata/extended amygdala, the posterior portion of the basal forebrain whose cholinergic neurons

project to the BLA and cortical mantle (*Rye et al., 1984*; *Zaborszky and Gyengesi, 2012*). Nearly 90% of neurons expressing ChR2 were cholinergic, as confirmed by immunostaining for choline acetyltransferase (ChAT), an obligate enzyme in the synthesis of ACh (*Prado et al., 2002 Figure 1—figure supplement 1A and B*). During early WoOT training, which did not yet involve photostimulation, mice were trained for a mean of seven sessions (*Figure 1—figure supplement 1C*), and both ChR2 and eYFP mice were similarly likely to lick at least once on an unsignaled *Window of Opportunity* as on an *Unrewarded Window*, confirming that mice could not predict reward delivery prior to licking, and that viral expression in the absence of photostimulation did not influence behavior (*Figure 1—figure supplement 1D*). Prior to photostimulation, ChR2 and eYFP groups also collected similar numbers of rewards (*Figure 1—figure supplement 1E*), had similar lick rates over entire sessions (*Figure 1—figure supplement 1F*), and had similar lick bout characteristics (*Figure 1—figure supplement 1G–I*), suggesting a stable setting in which to study the effects of transient photostimulation of cholinergic basal forebrain neurons.

## Transient photostimulation of basal forebrain cholinergic neurons increases conditioned responding when paired with the opportunity to collect rewards

After mice had undergone initial WoOT training, we were then able to investigate whether transient photostimulation of basal forebrain cholinergic neurons affected conditioned responding under different conditions (*Figure 1E–G*). We first tested whether photostimulation of basal forebrain cholinergic neurons inherently affected licking by delivering 2 s of photostimulation exclusively during *Unrewarded Windows*, that is, when licking would not be reinforced (Photostim-Unreinforced session, *Figure 1F*). In these Photostim-Unreinforced sessions, licking did not increase during photostimulated windows compared to windows with no photostimulation, in either ChR2 or eYFP mice (*Figure 1H*).

In separate sessions, we tested whether transient photostimulation of basal forebrain cholinergic neurons affected licking when photostimulation was delivered exclusively during a subset of unsignaled *Windows of Opportunity*, that is, when licking would be reinforced (*Figure 1G*, Photostim-Reinforced session, right panel). Given that there was no discrete external cue for these windows, we had initially hypothesized that cholinergic basal forebrain photostimulation would not impact licking behavior. However, in these Photostim-Reinforced sessions, photostimulation increased licking in mice expressing ChR2, but not eYFP (*Figure 1I*). In contrast, during windows with no photostimulation, licking remained similar between ChR2 and eYFP mice (*Figure 1I*). Transient photostimulation of basal forebrain cholinergic neurons became capable of promoting an operant conditioned response, increasing the likelihood and decreasing the latency of licking, but only after being paired with the opportunity to collect rewards.

While we initially used photostimulation parameters similar to prior work (*Herman et al., 2016*; *Jiang et al., 2016*), we also examined whether even briefer photostimulation of cholinergic basal forebrain neurons was sufficient to affect conditioned responding. In a separate session, we randomly delivered photostimulation on *Windows of Opportunity* using either 0, 1, 2, or 10 pulses at 20 Hz, corresponding to 5–500 ms of photostimulation (*Figure 1J–K*). Even a single 5 ms laser pulse was sufficient both to increase the likelihood of licking (*Figure 1J*) and accelerate the latency of the first lick following photostimulation onset (*Figure 1K*). In summary, brief photostimulation of cholinergic basal forebrain neurons can promote conditioned responding when paired with potential reinforcement. Further examples and analyses of the licking behavior across the Photostim-Unrewarded and Photostim-Rewarded sessions are depicted in *Figure 1—figure supplements 2 and 3*.

## Photostimulation of basal forebrain cholinergic neurons does not increase unconditioned movement and is not inherently reinforcing

Given that photostimulation of basal forebrain cholinergic neurons drove conditioned behavior when paired with potential reinforcement, we performed a series of experiments to determine whether the effects of photostimulation were specific to conditioned responding. We first examined whether photostimulation of cholinergic neurons increased other types of movement, for example, locomotion in an unreinforced context. Basal forebrain cholinergic photostimulation did not increase locomotion in an open-field test (OFT, *Figure 1—figure supplement 4A and B*), as measured by mouse velocity (*Figure 1—figure supplement 4C*). Additionally, photostimulation did not modify innate location

preference, as measured by the amount of time spent in the center of the open field, which was avoided similarly by ChR2 and eYFP mice (*Figure 1—figure supplement 4D*).

We next assessed whether photostimulation of cholinergic basal forebrain neurons inherently reinforced other types of locomotor behavior (*Figure 1—figure supplement 4E*). Mice were freely allowed to explore a chamber in which one half was paired with photostimulation, in a real-time place preference (RTPP) assay. Neither ChR2 nor eYFP mice demonstrated a preference for the side paired with photostimulation, suggesting that photostimulation of cholinergic basal forebrain neurons was not inherently reinforcing (*Figure 1—figure supplement 4F*).

Since cholinergic photostimulation did not increase movement in unreinforced contexts, we also tested whether cholinergic photostimulation affected unconditioned movements within our reinforced, behavioral task context. A subset of mice was trained while on a treadmill in order to measure spontaneous, unconditioned locomotion. There was no difference in locomotion around the time of photostimulation between ChR2 and eYFP mice during either Photostim-Unreinforced or Photostim-Reinforced sessions (*Figure 1—figure supplement 4G*). Locomotion also did not differ significantly between Photostim-Unreinforced and Photostim-Reinforced sessions for either ChR2 or eYFP mice (*Figure 1—figure supplement 4H*).

Lastly, we examined whether photostimulation of cholinergic basal forebrain neurons increased arousal during our head-fixed task, as measured by pupil diameter, using machine learning based pupillometry (DeepLabCut, *Figure 1—figure supplement 4I–L*). During Photostim-Unreinforced sessions, photostimulation of cholinergic neurons caused a modest increase in pupil dilation in ChR2 mice compared to eYFP mice (*Figure 1—figure supplement 4I*, left; rank-sum between groups, $p<0.05$). The first significant difference in pupil size between ChR2 and eYFP mice began 1.2 s after laser onset. During Photostim-Reinforced sessions, photostimulation also caused an increase in pupil dilation in ChR2 mice compared to eYFP mice, now beginning 0.7 s after laser onset (*Figure 1—figure supplement 4I*, right; rank-sum between groups, $p<0.05$). Pupil changes were larger during Photostim-Reinforced than Photostim-Unreinforced sessions for ChR2 mice (*Figure 1—figure supplement 4J*). There was no clear change in pupil size after photostimulation for eYFP mice (*Figure 1—figure supplement 4J*).

Some of the differences in photostimulation-evoked pupil dilation during the Photostim-Reinforced session for ChR2 mice may have been related to reward collection behavior. We examined pupillary diameter at the time of reward delivery in both Photostim-Unreinforced and Photostim-Reinforced sessions (*Figure 1—figure supplement 4K*). During both sessions, pupils dilated similarly in ChR2 and eYFP mice following reward delivery (*Figure 1—figure supplement 4L*). Therefore, cholinergic photostimulation increases arousal modestly, but this effect becomes evident more slowly for pupillary dilation than for licking behavior (0.5 s latency for licking as in *Figure 1*).

## Muscarinic receptors are necessary for conditioned responding

ACh can affect postsynaptic neurons in target regions through two classes of receptors: fast, ionotropic nicotinic receptors and relatively slower, G-protein coupled, metabotropic muscarinic receptors (*Brown, 2019*). In order to test which receptors mediated the effects of photostimulation of cholinergic basal forebrain neurons on conditioned responding, we blocked each receptor class using intraperitoneal injections of either a muscarinic antagonist (scopolamine) (*Marta et al., 2011*; *Kruk-Słomka et al., 2014*) or a nicotinic antagonist (mecamylamine) (*Adermark et al., 2014*; *Zachariou et al., 2001*; *Figure 2A*).

In order to understand the specific sensitivity of photostimulation-induced licking, however, and to increase within-subject control, we used a different task structure, in which mice were also trained to respond to tones (i.e., where a tone indicated a *Window of Opportunity*) (*Figure 2B*) prior to injections. On separate trials, mice received either a tone or transient photostimulation (2 s duration each). If mice licked after the onset of the tone or after the onset of photostimulation (within 3 s), they received a fluid reward. Licking after either tones or photostimulation was compared to licking on matched unrewarded windows during the ITI, a measure of baseline licking likelihoods. This more traditional operant task, using either tones or photostimulation, can be thought of as a variation of the WoOT operant task – in this version, there are no longer unsignaled windows: all windows are now potentially signaled by either tones or photostimulation, granting improved within subject control to assess the pharmacologic sensitivity of each trial type: tone, photostimulation, or baseline licking.

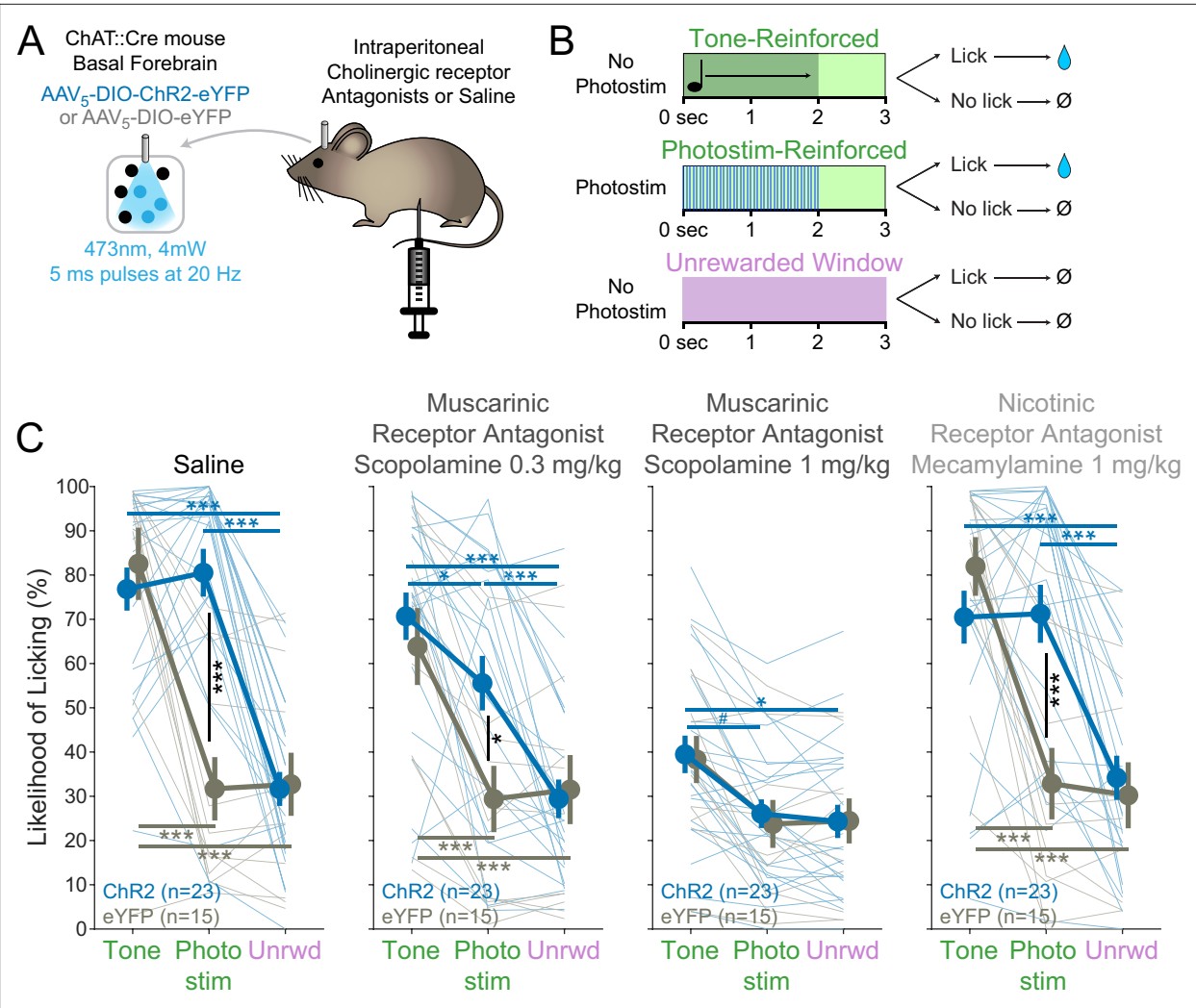

**Figure 2.** BF$^{ChAT}$:ChR2-induced conditioned responding is muscarinic receptor-dependent. (**A**) Experimental strategy to test the necessity of cholinergic receptors in conditioned responding. Cholinergic neurons in the basal forebrain were photostimulated after blockade of cholinergic muscarinic or nicotinic receptors using intraperitoneal injection of pharmacologic antagonists. (**B**) Modification of Window of Opportunity Task (WoOT) to include additional tone conditioned responses. To provide additional within-subject controls for pharmacological testing, mice could now receive rewards on either of two separate types of *Windows of Opportunity*: with tones (top row) or with photostimulation (middle row, both 2 s duration). Licks during unsignaled, *Unrewarded Windows* were recorded but had no consequence (bottom row). (**C**) Photostim-induced licking was abolished by systemic muscarinic receptor antagonist administration. Linear mixed effects modeling confirmed that licking depended upon an interaction between Virus group, Stimulus type, and Drug session ($F_{3,363}$ = 4.61, p=0.0002). Thin lines represent data from individual mice, pooled data are displayed as mean ± SEM. Saline: both ChR2 and eYFP mice responded more during Tone *Windows of Opportunity* than during unsignaled, *Unrewarded Windows* (Unrwd) (***p<0.001, *p<0.05, #p=0.10, Sidak post hoc multiple comparisons). However, only ChR2 mice responded more during Photostimulation *Windows of Opportunity* than during unsignaled, *Unrewarded Windows*, at a similar likelihood as their responses during tone *Windows of Opportunity*. Scopolamine 0.3 mg/kg: ChR2 mice now responded less during photostimulation than during tones. Scopolamine 1 mg/kg: ChR2 mice no longer responded more during photostimulation than during *Unrewarded Windows*, and no longer responded more during photostimulation than eYFP mice, although they continued to respond more during tones than intertrial intervals (ITIs). Mecamylamine 1 mg/kg: response patterns were similar to saline sessions. For each session, the likelihood of licking during *Unrewarded Windows* was similar between ChR2 and eYFP mice (all p>0.8). Additionally, within each group, the likelihood of licking during *Unrewarded Windows* was similar to saline sessions for all drug doses (all p>0.8).

The online version of this article includes the following source data for figure 2:

**Source data 1.** The likelihood of licking under saline and various agonist conditions and in response to different stimuli, as shown in *Figure 2C*.

On a control day in which mice were injected with saline, both ChR2 and eYFP mice responded more during tones than during ITIs (*Figure 2C*). However, only optogenetic ChR2 mice responded more during photostimulation than during ITIs, responding at similar rates during photostimulation as they did during tones.

When mice received an injection of the muscarinic receptor antagonist scopolamine (0.3 mg/kg), however, ChR2 mice began to lick less during photostimulation than tones (*Figure 2C*, middle panels). With a higher dose of scopolamine (1 mg/kg), ChR2 mice licked similarly on photostimulation trials as during ITIs, despite still licking more during tone trials than ITIs. Results from both doses suggested that blocking muscarinic receptors decreased conditioned responding (tones and photostimulation), but that conditioned responding to basal forebrain cholinergic neuron photostimulation was more sensitive to muscarinic blockade than conditioned responding to tones.

In contrast, conditioned responding to tones and photostimulation after injection of the nicotinic receptor antagonist mecamylamine was similar to that after saline control for both ChR2 and eYFP mice (*Figure 2C*, right). Therefore, muscarinic receptors, rather than nicotinic receptors, were necessary for conditioned responding to basal forebrain cholinergic neuron photostimulation.

## Basal forebrain cholinergic neurons are active during conditioned responding

Having determined that photostimulation of basal forebrain cholinergic neurons can promote conditioned responding even in the absence of discrete external cues, we next examined whether basal forebrain cholinergic neurons are inherently active during conditioned responding. Basal forebrain cholinergic neurons have been described to be active during reinforcer delivery (*Hangya et al., 2015*), reward consumption (*Harrison et al., 2016*), movements including locomotion (*Harrison et al., 2016*; *Nelson and Mooney, 2016*), and conditioned stimuli (*Guo et al., 2019*; *Parikh et al., 2007*). However, we observed that photostimulation of basal forebrain cholinergic neurons increased conditioned responding independent of these factors: in the absence of conditioned cues, prior to reinforcer delivery or consumption, and without affecting unconditioned movements such as locomotion. We therefore studied whether basal forebrain cholinergic neuron activity also changes at the time of conditioned responding in the absence of reward using a genetically encoded fluorescent calcium indicator, GCaMP6s (*Chen et al., 2013*), as a proxy for neural activity.

We targeted expression of GCaMP6s to basal forebrain cholinergic neurons using virally mediated, Cre-dependent expression (AAVdj-EF1a-DIO-GCaMP6s) (*Figure 3A*). An optic fiber implanted over the basal forebrain enabled real-time recording of fluctuations in neural activity using fiber photometry (*Adelsberger et al., 2005*; *Cui et al., 2013*; *Gunaydin et al., 2014*; *Lütcke et al., 2010*). We recorded GCaMP6s fluorescence while mice performed a more traditional operant task to detect tones. Licking after the onset of tones (within 3 s) was rewarded (*Figure 3B*). Recordings during this task demonstrated fluorescent transients in the 470 nm signal channel that appeared linked to behavioral events (*Figure 3C*). Peri-event analyses suggested that fluorescence levels increased at the time of behavioral events, most clearly with licking in the presence of a tone (*Figure 3D and E*), but also during spontaneous licking (i.e., in the absence of tone cues and reward delivery) (*Figure 3F and G*). There were no apparently meaningful changes in the 405 nm reference channel, suggesting that changes in fluorescence were not related to simple movement artifacts.

It is possible that basal forebrain cholinergic neuron activity during licking even in the absence of tone cues was in some way influenced by the broader cue-reinforcer association context of this task. To control for this, we also recorded fluorescent activity from basal forebrain cholinergic neurons prior to any experience with tone cues, during our uncued reward task (WoOT, *Figure 3H*). Even in this context, without discrete tones or cues, changes in fluorescent activity were observed both at the time of licking that triggered reward delivery (*Figure 3I*), as well as at the time of licking in the absence of reward delivery (*Figure 3J–K*). This suggests that cholinergic neurons are physiologically active at the time of conditioned responding, even in the absence of cues and reward consumption.

## Local ACh levels in the BLA, measured using a genetically encoded sensor, increase during conditioned responses

Although we had determined that basal forebrain cholinergic neuron activity increases with conditioned responding (*Figure 3*), we wanted to confirm whether ACh is released at these times into target

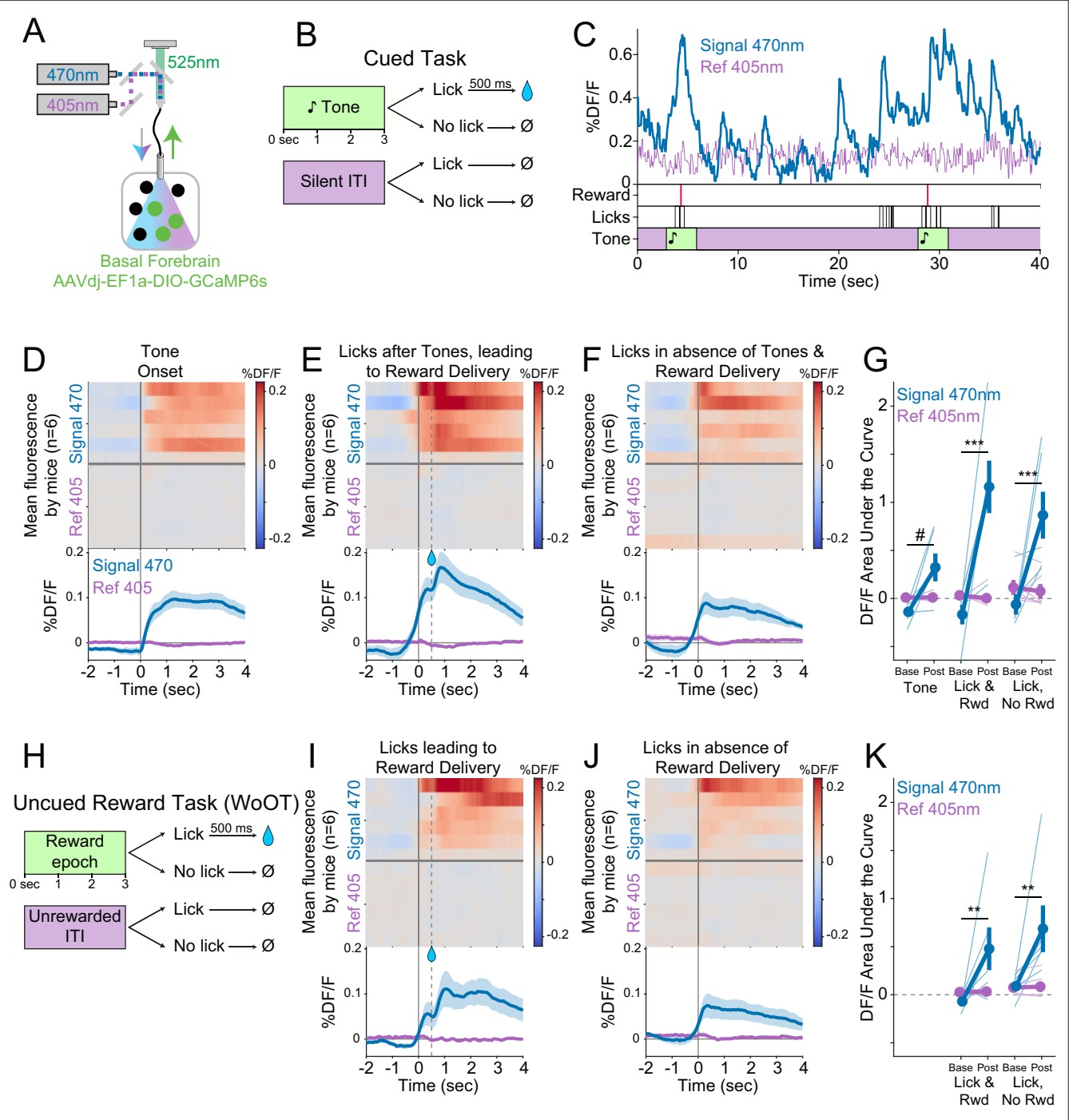

**Figure 3.** Cholinergic basal forebrain neural activity increases during conditioned stimuli and responses, even in the absence of reward delivery.
(**A**) Strategy to record fluorescent activity from basal forebrain cholinergic neurons expressing the calcium sensor GCaMP6s, using interleaved signal (470 nm, blue) and reference (405 nm, violet) wavelengths to elicit fluorescence (525 nm, green). (**B**) Task windows. We recorded fluorescent activity from mice during a traditional operant cue detection task. If mice licked after the onset of a tone, a fluid reward was delivered after a 0.5 s delay. Licks during the silent *Unrewarded Windows* had no consequence. (**C**) Sample GCaMP photometry fluorescence traces from one mouse demonstrating signal increases around the times of tones, licks, and reward deliveries. Increases were apparently present even for licking in the absence of tones and rewards. The blue trace represents data from the signal wavelength (470 nm) and the violet trace represents interleaved data from the reference wavelength (405 nm). (**D**) Changes in fluorescence from basal forebrain cholinergic neurons referenced to the time of tone onset. Heat maps represent trial-averaged data from each mouse. Top heat maps are for 470 nm excited fluorescence (Signal 470), bottom heat maps are for 405 nm reference (Ref 405). The bottom panel summary data are represented as mean ± SEM. Mice are sorted in all heat maps (**D–F, I–J**), in the order of average post-lick activity in panel (**E**). (**E**) Changes in fluorescence from basal forebrain cholinergic neurons referenced to the time of the first lick after tone onset. Licking

*Figure 3 continued on next page*

*Figure 3 continued*

triggered subsequent reward delivery (first lick at 0 s, reward delivery at dashed line, 0.5 s). (**F**) Changes in fluorescence from basal forebrain cholinergic neurons referenced to the onset of matched lick bouts that were in the absence of tone cues and did not lead to reward delivery. (**G**) Fluorescence levels from basal forebrain cholinergic neurons at baseline (−2 to −1.5 s before each referenced event) and post-event time points (0–0.5 s after events, to standardize analyses between events) in the cued task. Fluorescence levels depended on an interaction of wavelength, time point, and event type (linear mixed effects model, $F_{2,55} = 3.28$, $p=0.045$). Fluorescence levels increased at the time of events in the 470 nm wavelength signal channel (blue), but not the 405 reference channel (violet) (tone: $t_{55} = 2.64$, #$p=0.064$; Lick leading to Reward Delivery [Lick & Rwd]: $t_{55} = 7.57$, ***$p<0.001$; Licks in the absence of Cue & Reward Delivery [Lick, No Rwd]: $t_{55} = 5.29$, ***$p<0.001$; Sidak correction for six multiple comparisons). Thin lines represent data from all individual mice, pooled data are represented as mean ± SEM. (**H**) Uncued Window of Opportunity Task (WoOT). All mice were also recorded from at an earlier stage of WoOT training, before experience with tones or other discrete cues. If a mouse licked during an uncued *Window of Opportunity*, a fluid reward was delivered. A 0.5 s delay was instituted between lick and reward to account for the slow dynamics of GCaMP6s. Licks during *Unrewarded Windows* were recorded but had no consequence. (**I**) Changes in fluorescence from basal forebrain cholinergic neurons referenced to the time of the first lick that triggered reward delivery (first lick at 0 s, reward delivery at dashed line, 0.5 s). (**J**) Changes in fluorescence from basal forebrain cholinergic neurons referenced to the onset of matched lick bouts that did not lead to reward delivery. (**K**) Fluorescence levels from basal forebrain cholinergic neurons at baseline (−2 to −1.5 s before each referenced event) and post-event time points (0–0.5 s after events, to standardize analyses between events) in WoOT. Fluorescence levels depended on an interaction of wavelength and time point (linear mixed effects model, $F_{1,35} = 13.59$, $p=0.0008$), without a third-order interaction by event type ($F_{1,35} = 0.02$, $p=0.882$). Fluorescence levels increased at the time of events in the 470 nm wavelength signal channel (blue), but not the 405 reference channel (violet) (Lick leading to Reward Delivery [Lick & Rwd]: $t_{35} = 3.58$, **$p=0.004$; Licks in absence of Reward Delivery [Lick, No Rwd]: $t_{35} = 3.90$, **$p<0.002$; Sidak correction for four multiple comparisons). Thin lines represent data from all individual mice, pooled data are represented as mean ± SEM.

The online version of this article includes the following source data for figure 3:

**Source data 1.** The fluorescence levels during baseline and post-cue events in the cued task, as shown in *Figure 3G*.

**Source data 2.** The fluorescence levels during baseline and post-cue events in WoOT, as shown in *Figure 3K*.

regions. We measured ACh release within the BLA using a novel version of a genetically encoded ACh sensor (GRAB$_{ACh3.0}$, abbreviated as GACh3.0 hereafter) (*Jing et al., 2018*), whose fluorescence reports the dynamics of extracellular ACh (*Figure 4A and B*), with somewhat faster kinetics than GCaMP6s (*Chen et al., 2013*).

We drove expression of the genetically encoded ACh sensor in BLA neurons by injecting an adeno-associated virus carrying the ACh sensor (AAV$_9$-hSyn-GACh3.0) into the BLA (*Figure 4A*). An optic fiber implanted over the BLA enabled real-time recording of ACh dynamics in the BLA by fiber photometry. We recorded local ACh while mice performed a more traditional operant task to detect tones, during which licking after tone onset was rewarded (*Figure 4C*). Sample recordings during this task revealed fluorescent transients that appeared linked to behavioral events (*Figure 4D*). Similar to somatic GCaMP photometry, GACh3.0 fluorescence signals increased at the time of behavioral events, significantly for licking leading to reward delivery (*Figure 4E and F*). We also recorded BLA ACh dynamics prior to any experience with tone cues, during our uncued reward task (WoOT, *Figure 4I*). Even in this context, without discrete tones or other discrete cues, changes in fluorescent signals were observed both at the time of licking that triggered reward delivery and at the time of licking without subsequent reward delivery (*Figure 4J–L*). These results suggest that BLA ACh levels increased at the time of conditioned responding, even in the absence of discrete cues and reward consumption.

## Photostimulation of basal forebrain cholinergic terminals in the BLA increases conditioned behavior when paired with reinforcement

Having confirmed that ACh levels in the BLA increase at the time of conditioned responding, we next studied whether photostimulation of basal forebrain cholinergic terminals directly within the BLA was sufficient to promote conditioned responding. Using Cre-dependent targeting, we again expressed either ChR2 or a control fluorophore (eYFP) in basal forebrain cholinergic neurons (*Figure 5A*). To stimulate cholinergic terminals in the BLA, we placed an optic fiber over the BLA.

Following initial uncued training, we tested whether transient photostimulation of basal forebrain cholinergic terminals in the BLA affected conditioned responding under two different conditions. When photostimulation was performed during *Unrewarded Windows* (Photostim-Unreinforced sessions), licking did not increase compared to statistically matched baseline windows in either ChR2 or eYFP mice (*Figure 5B*). When photostimulation was delivered exclusively during a subset of *Windows of Opportunity* (Photostim-Reinforced sessions), photostimulation increased licking only in mice expressing ChR2 (*Figure 5C*). Baseline licking rates, however, remained similar between ChR2

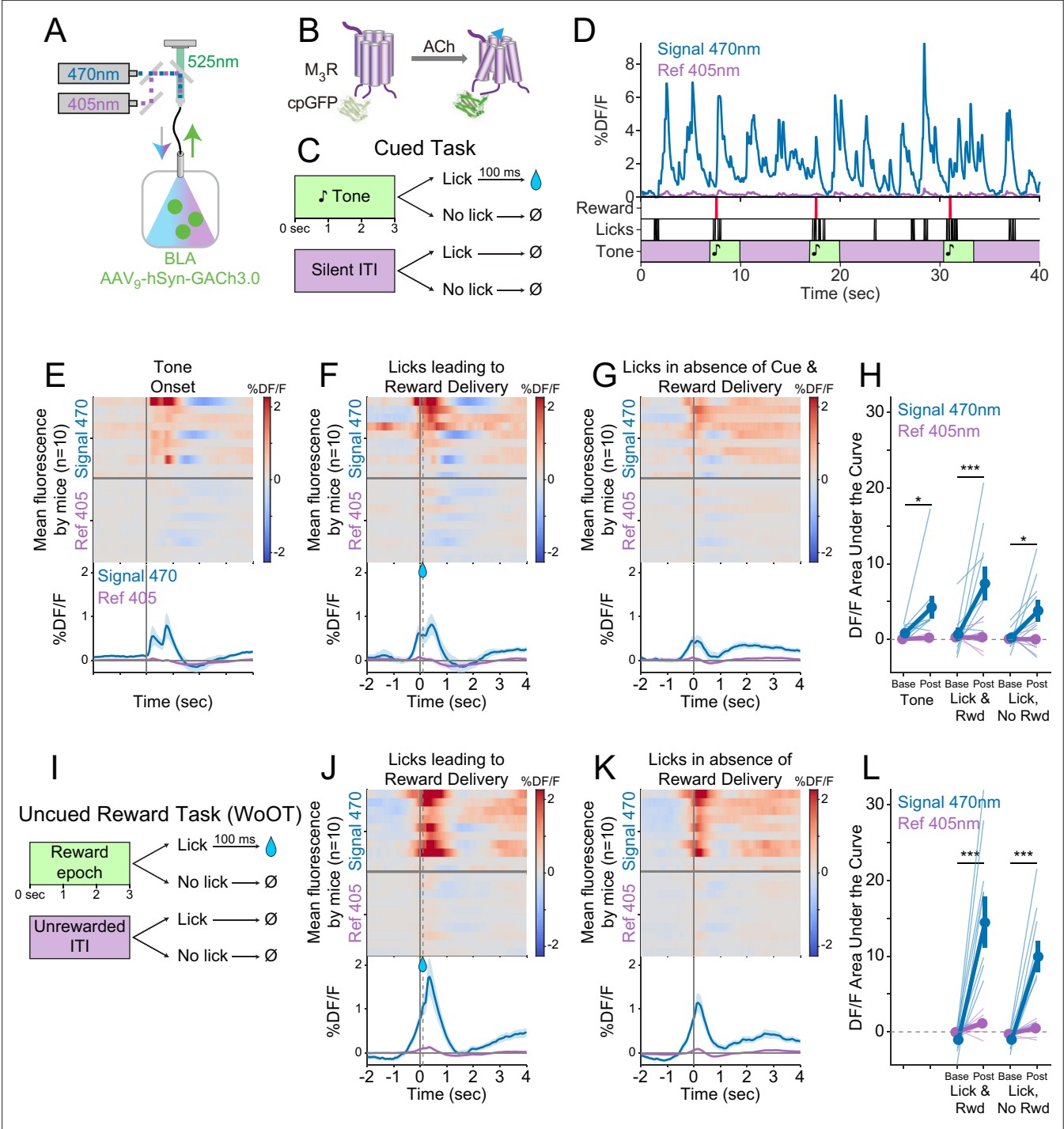

**Figure 4.** Local acetylcholine (ACh) levels in the basolateral amygdala (BLA), measured using a genetically encoded sensor, increase during conditioned stimuli and responses. (**A**) Strategy to record local ACh levels in the BLA. A genetically encoded, fluorescent ACh sensor (GACh3.0, [**B**]) was expressed in BLA neurons, and imaged using interleaved signal (470 nm, blue) and reference (405 nm, violet) wavelengths to elicit fluorescence (525 nm, green). (**B**) The fluorescent ACh sensor, GACh3.0, is a fusion protein between a modified M3 muscarinic receptor and cyclically permuted GFP. GACh3.0 undergoes a conformational change and fluoresces to 470 nm light after binding ACh. Please note that kinetics for GACh3.0 (*Jing et al., 2020*) are somewhat faster than those for GCaMP6s.(*Chen et al., 2013*). (**C**) Cued task windows. We recorded fluorescent activity from mice during a traditional operant cue detection task. If mice licked after the onset of a tone, a fluid reward was delivered after a 0.1 s delay. Licks during intertrial intervals (ITIs) had no consequence. (**D**) Sample BLA ACh sensor fluorescence traces from one mouse demonstrating apparent increases around the times of tones, licks, and reward deliveries. Increases were apparently present even for licking in the absence of tones and rewards. (**E**) Changes in BLA ACh sensor fluorescence referenced to the time of tone onset. Heat maps represent trial-averaged data from each mouse. Top heat maps are for 470 nm excited

*Figure 4 continued on next page*

*Figure 4 continued*

fluorescence (Signal 470), bottom heat maps are for 405 nm reference (Ref 405). The bottom panel summary data are represented as mean ± SEM. Mice are sorted in all heat maps (**E–G, J–K**), in the order of average post-lick activity in panel (**F**). (**F**) Changes in BLA ACh sensor fluorescence referenced to the time of the first lick after tone onset. Licking triggered reward delivery (first lick at 0 s, reward delivery at dashed line, 0.5 s). (**G**) Changes in BLA ACh sensor fluorescence referenced to the onset of matched lick bouts that were in the absence of tone cues and did not lead to reward delivery. (**H**) BLA ACh sensor fluorescence levels at baseline (−2 to −1.5 s before each referenced event) and post-event time points (0–0.5 s after events, to standardize analyses between events) in the Cued task. Fluorescence levels depended on an interaction of wavelength and time point (linear mixed effects model, $F_{1,99}$ = 20.41, p<0.001), without a third-order interaction by event type ($F_{2,99}$=1.20, p=0.305). Fluorescence levels increased at the time of events in the 470 nm wavelength signal channel (blue), but not the 405 reference channel (violet) (tone: $t_{99}$ = 2.80, *p=0.036; Lick leading to Reward Delivery [Lick & Rwd]: $t_{99}$ = 5.49, ***p<0.001; Licks in the absence of Cue & Reward Delivery [Lick, No Rwd]: $t_{99}$ = 2.91, *p<0.027; Sidak correction for six multiple comparisons). Thin lines represent data from all individual mice, pooled data are represented as mean ± SEM. (**I**) Uncued Window of Opportunity Task (WoOT). All mice were also recorded from at an earlier stage of WoOT training, before experience with tones or other discrete cues. If a mouse licked during an uncued *Window of Opportunity*, a fluid reward was delivered. Licks during *Unrewarded Windows* were recorded but had no consequence. (**J**) Changes in BLA ACh sensor fluorescence referenced to the time of the first lick that triggered reward delivery (first lick at 0 s, reward delivery at dashed line, 0.1 s). (**K**) Changes in BLA ACh sensor fluorescence referenced to the onset of matched lick bouts that did not lead to reward delivery. (**L**) BLA ACh sensor fluorescence levels at baseline (−2 to −1.5 s before each referenced event) and post-event time points (0–0.5 s after events, to standardize analyses between events) in WoOT. Fluorescence levels depended on an interaction of wavelength and time point (linear mixed effects model, $F_{1,63}$ = 44.21, p<0.001), without a third-order interaction by event type ($F_{1,63}$ = 1.25, p=0.267). Fluorescence levels increased at the time of events in the 470 nm wavelength signal channel (blue), but not the 405 reference channel (violet) (Lick leading to Reward Delivery [Lick & Rwd]: $t_{63}$ = 8.41, ***p<0.001; Licks in the absence of Reward Delivery [Lick, No Rwd]: $t_{63}$ = 5.94, ***p<0.001; Sidak correction for four multiple comparisons). Thin lines represent data from all individual mice, pooled data are represented as mean ± SEM.

The online version of this article includes the following source data for figure 4:

**Source data 1.** BLA ACh sensor fluorescence levels at baseline and post-event time points in the Cued task, as shown in *Figure 4H*.

**Source data 2.** BLA ACh sensor fluorescence levels at baseline and post-event time points in the WoOT, as shown in *Figure 4L*.

and eYFP mice. Hence, transient photostimulation of basal forebrain cholinergic terminals in the BLA drove conditioned responding in a temporally precise way, only when photostimulation was associated with the opportunity to collect rewards, similar to results with somatic basal forebrain cholinergic neuron photostimulation (*Figure 1*).

## Concurrent photostimulation of cholinergic terminals and local ACh measurement in the BLA reveals that levels of photo-elicited ACh do not change when associated with reward

Why should photostimulation of cholinergic basal forebrain neurons lead to conditioned responding when associated with reinforcement, but not when unassociated? We postulated that either (1) reinforcement potentiates the amount of ACh released with photostimulation (a 'presynaptic' explanation) or (2) the amount of ACh released by photostimulation does not change, but the effects on downstream neurons are modified by reinforcement (a 'postsynaptic' explanation). In order to determine whether the amount of ACh released by photostimulation changes as a result of pairing with reinforcement, we measured local ACh in the BLA while concurrently photostimulating basal forebrain cholinergic terminals (*Figure 5D*). We placed an optic fiber over the BLA in order to use orange light (589 nm) to transiently photostimulate basal forebrain cholinergic terminals using a Cre-dependent, red-shifted optogenetic protein (ChrimsonR) expressed in the basal forebrain of ChAT::Cre mice. We concurrently shined blue light (470 nm) through the same optic fiber to measure local ACh levels using a genetically encoded ACh sensor in BLA neurons. Sample recordings demonstrated that ACh sensor fluorescence increased during photostimulation only in mice expressing ChrimsonR (*Figure 5E*), but not in those expressing a control fluorophore (tdTomato, *Figure 5F*).

We compared, in the same mice, BLA levels of ACh fluorescence during photostimulation during Photostim-Unreinforced and Photostim-Reinforced sessions (*Figure 5G–I*). Photostimulation elicited robust responses in mice expressing ChrimsonR, but not in mice expressing a control fluorophore (*Figure 5G*). The amount of measured fluorescence did not depend upon whether photostimulation was provided in Photostim-Unreinforced or Photostim-Reinforced sessions. Taken together with our previous finding that photostimulation of cholinergic terminals in the BLA selectively elevated licking during Photostim-Reinforced sessions (*Figure 5C*), our observation that BLA ACh levels did not

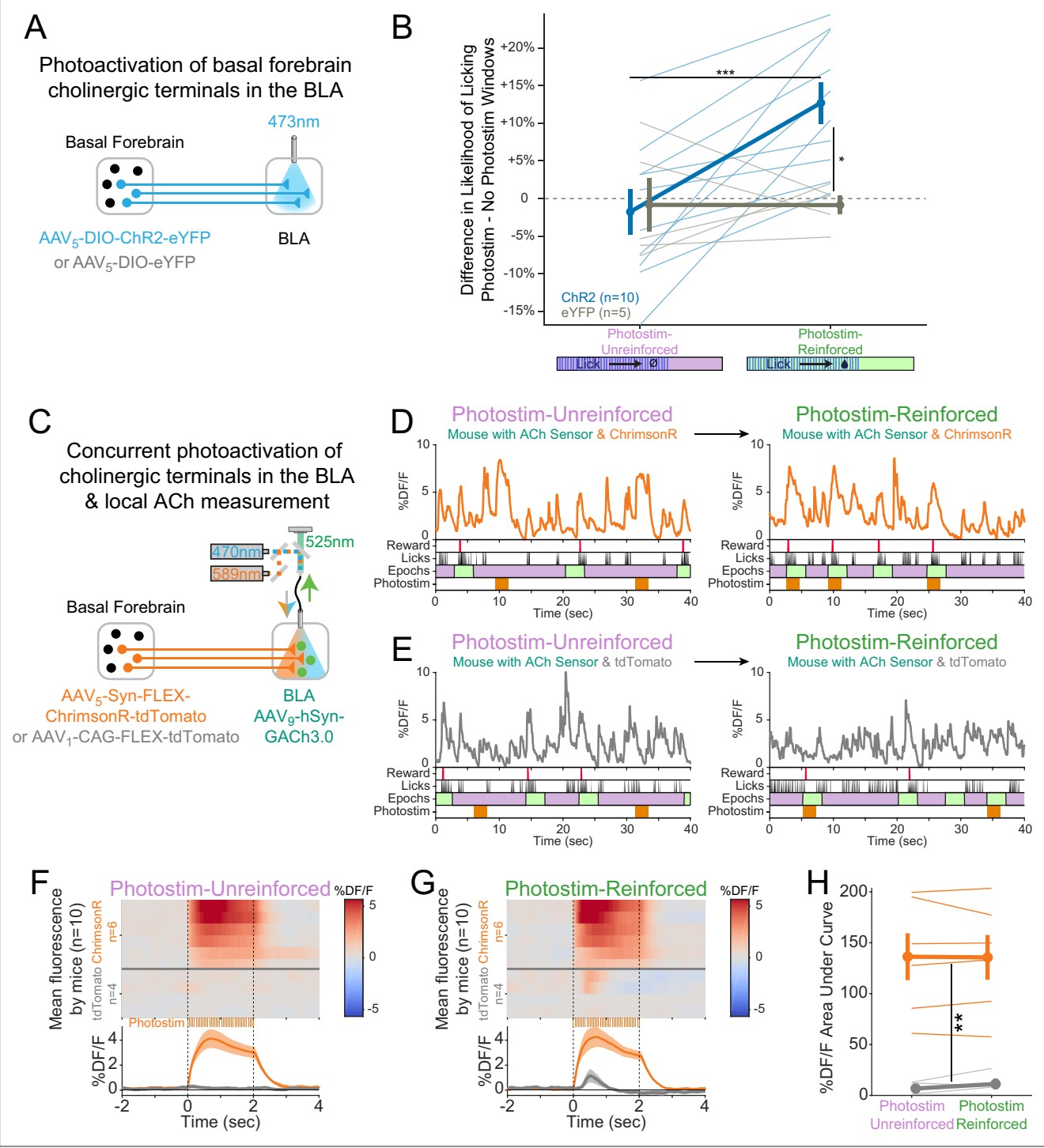

**Figure 5.** Cholinergic signaling in the basolateral amygdala (BLA) is sufficient to promote conditioning responding but acetylcholine (ACh) release is independent of reward contingency. (**A**) Optogenetic strategy to photostimulate cholinergic (ChAT::Cre) basal forebrain terminals in the BLA selectively. (**B**) The difference in the likelihood of licking between Photostim and No Photostim windows differed depending on the window in which photostimulation was delivered. The effect of photostimulation within each session is calculated for each mouse. A linear mixed effects model to account for repeated measures demonstrated that the effect of photostimulation depended on an interaction between Virus and Session type ($F_{1,15}$ = 9.624, p=0.007). Post hoc tests Sidak correction for multiple comparisons revealed that the effect of photostimulation was greater for ChR2 mice in Photostim-Reinforced than Photostim-Unreinforced sessions (***p=0.0004), and that the effect of photostimulation in Photostim-Reinforced sessions was greater for ChR2 mice than eYFP mice (*p=0.0215). All other comparisons were not significant (p>0.05). (**C**) Schematic showing concurrent photostimulation of cholinergic terminals in the BLA while measuring local ACh using a genetically encoded fluorescent sensor, through the same optic

*Figure 5 continued on next page*

*Figure 5 continued*

fiber. Mice either expressed ChrimsonR or a control fluorophore (tdTomato) in basal forebrain ChAT neurons.(**D**) Sample fluorescent traces from ACh sensor (orange) from a mouse with ChrimsonR, in relationship to reward delivery (red), licks (black), behavioral windows (Reward green/Unrewarded purple), and photostimulation (orange). Photostimulation was either delivered in a Photostim-Unreinforced session (during *Unrewarded Windows*, left, purple) or Photostim-Reinforced session (during rewarded windows, right, green). (**E**) Sample ACh fluorescent traces from ACh sensor (gray) from a mouse with a control fluorophore, displayed similarly to (**E**), in relationship to rewards, licks, behavioral windows, and photostimulation, delivered either in a Photostim-Unreinforced (left, purple) or Photostim-Reinforced session (right, green). (**F**) Heat maps comparing average ACh measurements for each mouse around the time of photostimulation on the Photostim-Unreinforced session. Mice are separated based on whether they expressed ChrimsonR (orange, n = 6) or control fluorophore (gray, n = 4). Summary data in the bottom panel are represented as mean ± SEM. Mice are sorted in all panels based mean DF/F during laser stimulation. (**G**) Heat maps comparing average ACh measurements for each mouse during photostimulation in the Photostim-Reinforced session. Conventions are as in (**G**), and mice are sorted in the same order as in (**G**). (**H**) Mean ACh measurements evoked by photostimulation on the Unreinforced (left) or Reinforced (right) session. Evoked ACh measurements were higher for ChrimsonR mice than control fluorophore mice, but evoked ACh measurements did not depend upon whether photostimulation was provided on Unreinforced or Reinforced sessions (linear mixed effects model: effect of Virus $F_{1,8}$ = 20.21, \*\*p=0.002; effect of Session $F_{1,8}$ = 0.47, p=0.51; interaction between Virus and Session type $F_{1,8}$ = 0.86, p=0.38).

The online version of this article includes the following source data and figure supplement(s) for figure 5:

**Source data 1.** The likelihood of licking for Photostim-Reinforced or Photostim-Unreinforced sessions, as shown in *Figure 5B*.

**Source data 2.** ACh measurements evoked by photostimulation for Photostim-Reinforced or Photostim-Unreinforced sessions, as shown in *Figure 5H*.

**Figure supplement 1.** Comparison of acetylcholine (ACh) evoked by reward collection with that evoked by photostimulation.

**Figure supplement 1—source data 1.** The fluorescence levels in response to Reward under various photostim durations, as shown in *Figure 5-figure supplement 1C*.

---

change with reward suggests that reward associations may gate postsynaptic responses to photostimulation, rather than presynaptically changing ACh release.

Lastly in these experiments, given that even briefer photostimulation had been capable of promoting conditioned responding (*Figure 1J and K*), we additionally performed recordings during a Photostim-Reinforced session, in which photostimulation was provided at various numbers of laser pulses at the same duty cycle. The amount of ACh sensor fluorescence evoked by 1–2 laser pulses, which had been sufficient to promote conditioned responding when photostimulation was targeted to the soma (*Figure 1*), was similar to levels of ACh at the time of reward delivery in the absence of any photostimulation (*Figure 5—figure supplement 1*). Therefore, a level of optogenetically induced ACh release similar to physiologic release can promote conditioned responding, specifically when paired with the opportunity to collect rewards (Photostim-Reinforced sessions).

## Cholinergic effects in vivo differ between target regions and depend upon reinforcer context in the amygdala

Because reinforcement context did not change the photostimulation elicited presynaptic release of ACh but led to increased conditioned responding, we next evaluated whether photostimulation effects on target regions involved in conditioned responding might depend on reinforcer context. The BLA and dorsomedial prefrontal cortex (dmPFC) are both involved in conditioned behavior (*Cardinal et al., 2002*), receive projections from basal forebrain cholinergic neurons (*Kitt et al., 1994*; *Woolf et al., 1984*), and are functionally interrelated (*Burgos-Robles et al., 2017*; *Likhtik et al., 2014*; *Likhtik et al., 2005*). We therefore studied how photostimulation of basal forebrain cholinergic neurons affects in vivo neural activity in these target regions (*Figure 6*). We again performed surgery on mice to express either ChR2 or a control fluorophore (eYFP) in basal forebrain cholinergic neurons using Cre-dependent targeting (*Figure 6A*). In addition to implanting optic fibers over the basal forebrain to photostimulate basal forebrain cholinergic neurons, we also implanted microwire bundles in the dmPFC and BLA to record single-unit activity in vivo during photostimulation.

During photostimulation of basal forebrain cholinergic neurons, dmPFC neural activity increased overall across the population (*Figure 6B*, left; *Figure 6—figure supplement 1*). The increase in dmPFC population neural activity during photostimulation was similar for both Photostim-Unreinforced sessions (purple) and Photostim-Reinforced sessions (green). In contrast to the dmPFC, the effects of photostimulation of basal forebrain cholinergic neurons on BLA neural activity differed depending on the session (*Figure 6B*, right). In Photostim-Unreinforced sessions (purple), the overall BLA population response consisted of facilitation, particularly at onset, whereas in the Photostim-Reinforced sessions

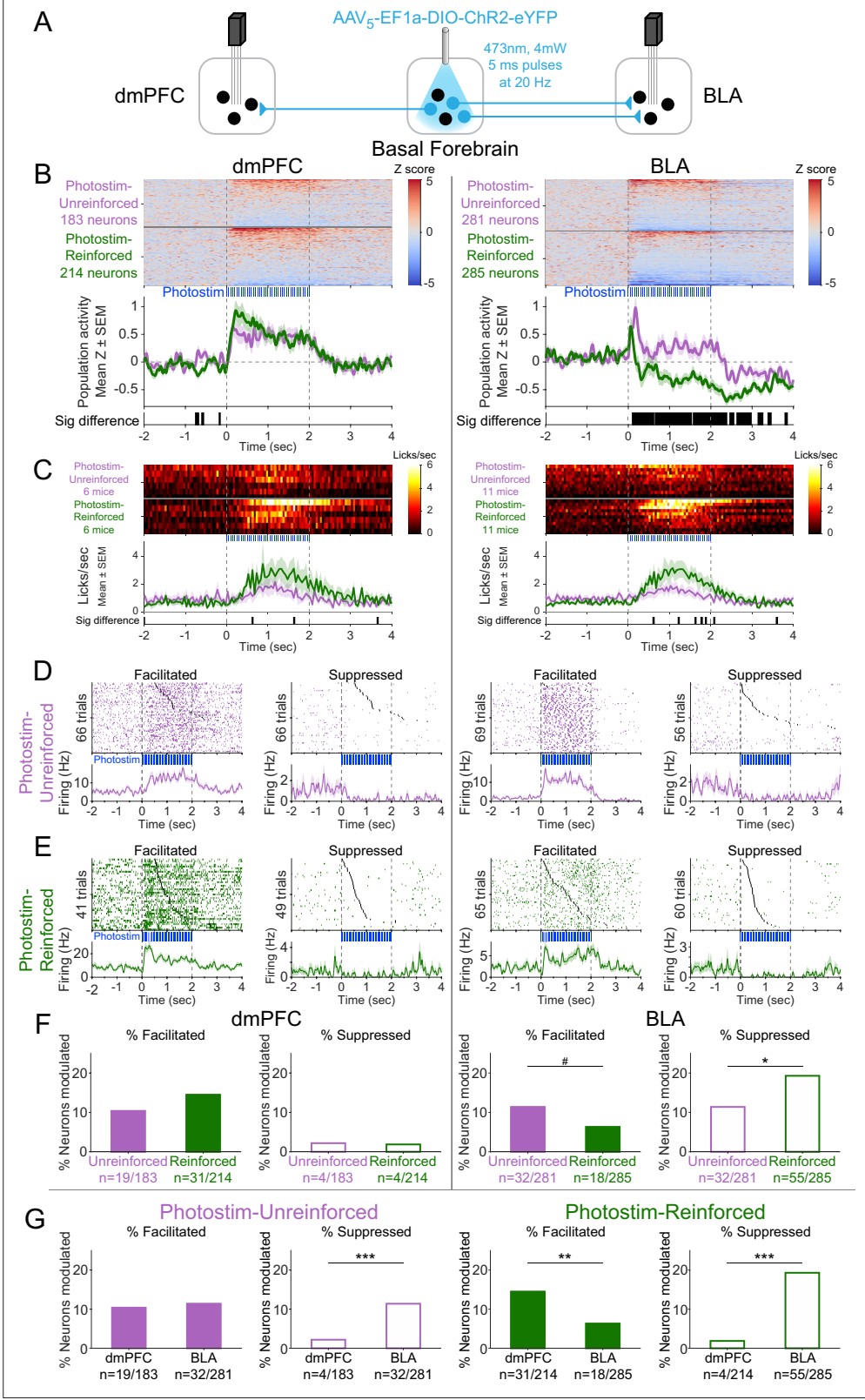

**Figure 6.** Cholinergic modulation of neural activity in vivo depends upon reward context in the amygdala, but not in the prefrontal cortex. (**A**) Strategy for photostimulation of cholinergic basal forebrain neurons and terminal region electrophysiology in the dorsomedial prefrontal cortex (dmPFC) and basolateral amygdala (BLA). Six ChR2 mice had electrodes implanted in both dmPFC and BLA. Five ChR2 mice had electrodes implanted only in the

*Figure 6 continued on next page*

*Figure 6 continued*

basolateral amygdala (BLA), yielding a total of 11 ChR2 mice with electrodes in BLA. Photostimulation parameters were the same as in ChR2 behavioral experiments (*Figure 1*). (**B**) Activity from all recorded neurons in each target area (dmPFC and BLA, total 963 neurons over all sessions from 11 mice), from sessions in which photostimulation was delivered during intertrial interval (ITI) *Unrewarded Windows* (Photostim-Unreinforced, purple) or sessions in which photostimulation was delivered during *Windows of Opportunity* (Photostim-Reinforced, green). Each row represents activity from a single neuron, normalized to baseline (−2 to 0 s before photostimulation onset) and smoothed with a 50 ms Gaussian. Neurons are sorted according to mean activity during photostimulation (0 to +2 s). Summary population data in the bottom panels are represented as mean ± SEM. Black marks underneath the population data represent 10 ms steps when the population activity differed between Photostim-Unreinforced vs. Photostim-Reinforced sessions (rank-sum test, p<0.01). (**C**) Licking activity from all mice contributing recordings for each target area, from sessions in which photostimulation was delivered during ITI *Unrewarded Windows* (Photostim-Unreinforced, purple) or sessions in which photostimulation was delivered during *Windows of Opportunity* (Photostim-Reinforced, green). Each row represents activity from a single mouse. Summary population data in the bottom panels are represented as mean ± SEM. Black marks underneath the population data represent 10 ms steps when the population licking activity differed between Photostim-Unreinforced vs. Photostim-Reinforced sessions (rank-sum test, p<0.01). (**D**) Example neural activity from each target area (dmPFC left, BLA right) around photostimulation of basal forebrain cholinergic neurons (0–2 s), from Photostim-Unreinforced sessions. Top panels are individual trial rasters and black markers indicate the first lick following Photostim onset. Trials are sorted by lick latency. Summary data in the bottom panels are represented as mean ± SEM, smoothed with a 50 ms Gaussian kernel. We observed neurons that were facilitated and suppressed relative to baseline in both regions (signed-rank test of firing rate in the 1 s before stimulation vs 0.5 s after, p<0.01). (**E**) Example neural activity from each target area (dmPFC left, BLA right) around photostimulation of basal forebrain cholinergic neurons (0–2 s) from Photostim-Reinforced sessions. We again observed neurons that were facilitated and suppressed relative to baseline in both regions. Conventions are the same as in (**E**). (**F**) Proportions of neurons that were facilitated (solid bars) or suppressed (open bars) in each area during Photostim-Unreinforced sessions (purple) or Photostim-Reinforced sessions (green). Denominator ns refer to neurons recorded across all mice during each session type. A higher percentage of BLA neurons were suppressed on Photostim-Reinforced sessions than Photostim-Unreinforced sessions (two-sample tests for equality of proportions: $X^2 = 6.81$, df = 1, *p=0.036, corrected for four multiple comparisons using Holm's procedure). There was a trend toward a lower percentage of BLA neurons being facilitated on Photostim-Reinforced sessions than Photostim-Unreinforced sessions ($X^2 = 4.52$, df = 1, #p=0.10). (**G**) Proportions of neurons that were facilitated (solid bars) or suppressed (open bars) in each area during Photostim-Unreinforced sessions (left, purple) or Photostim-Reinforced sessions (right, green). Data is replotted from (**E**) to facilitate comparisons between areas for each session type. A higher percentage of BLA neurons than dmPFC neurons were suppressed during both Photostim-Unreinforced and Photostim-Reinforced sessions (two-sample tests for equality of proportions: Photostim-Unreinforced: $X^2 = 13.11$, df = 1, ***p<0.001; Photostim-Reinforced: $X^2 = 35.61$, df = 1, ***p<0.001; all p values corrected for four multiple comparisons using Holm's procedure). A lower percentage of BLA neurons than PFC neurons was facilitated during Photostim-Reinforced sessions ($X^2 = 9.21$, df = 1, **p=0.005).

The online version of this article includes the following source data and figure supplement(s) for figure 6:

**Source data 1.** The neurons that were modulated (either facilitated or suppressed) by Photostim-Reinforced or Photostim-Unreinforced sessions for each brain region, as shown in *Figure 6F, G*.

**Figure supplement 1.** Example neural variability and relationship to behavioral responses.

**Figure supplement 2.** Responses of neurons in the dorsomedial prefrontal cortex (dmPFC) and basolateral amygdala (BLA) to shorter photostimulations.

**Figure supplement 3.** Effects of laser illumination on neural activity were not seen in control eYFP subjects.

**Figure supplement 3—source data 1.** The neurons that were modulated (either facilitated or suppressed) by Photostim-Reinforced or Photostim-Unreinforced sessions for each brain region for control eYFP and experimental ChR2 mice, as shown in *Figure 6-figure supplement 3D, E*.

**Figure supplement 4.** Neurons facilitated or suppressed by cholinergic photostimulation may have different baseline firing rates.

**Figure supplement 4—source data 1.** The baseline firing rate of neurons in each brain region of ChR2 mice that were modulated by photostimulation either in the Photostim-Reinforced or Photostim-Unreinforced sessions, as shown in *Figure 6-figure supplement 4A*.

(green), the BLA population response had a more striking and sustained suppression. The difference in the BLA population response between the two sessions was evident as early as 135 ms after photostimulation onset and occurred with even shorter durations of photostimulation (*Figure 6—figure supplement 2*). There were minimal changes related to photostimulation in control mice expressing eYFP for both Photostim-Unreinforced and Paired sessions types (*Figure 6—figure supplement 3*).

The population neural responses in the dmPFC and BLA reflected a mix of individual neurons that were facilitated or suppressed by cholinergic basal forebrain photostimulation (*Figure 6D and E*). In the dmPFC, the proportion of neurons facilitated and suppressed by photostimulation was stable across session types (*Figure 6F*, left). In the BLA, however, the proportion of individual neurons that were facilitated tended to decreased during Photostim-Reinforced sessions and the proportion of neurons that were suppressed increased during Photostim-Reinforced sessions (*Figure 6F*, right). Comparing between these regions (*Figure 6G*), a higher proportion of BLA neurons was suppressed than dmPFC neurons during both session types, but a lower proportion of BLA neurons than dmPFC neurons was facilitated during the Photostim-Reinforced sessions. These findings suggested that cholinergic effects in vivo differ between target regions and depend upon reinforcer context in the amygdala.

There was a striking heterogeneity of neural responses within each target region, with some neurons in each region facilitated and others suppressed. We therefore explored how neurons that were facilitated or suppressed might differ. Neurons in the dmPFC that were facilitated by photostimulation of cholinergic basal forebrain neurons during the Photostim-Reinforced sessions had higher baseline firing rates than neurons that were suppressed (*Figure 6—figure supplement 4*). There was a possible similar trend in the BLA ($p=0.057$ after correction for multiple comparisons) – neurons that were facilitated during the Photostim-Reinforced sessions had higher baseline firing rates, while neurons that were suppressed had lower baseline firing rates. The possible differences in firing rates suggested that different neurons might be facilitated or suppressed. To identify specific neural subpopulations of BLA neurons that might differentially respond to photostimulation of cholinergic basal forebrain neurons, we transitioned to an ex vivo preparation.

## Cholinergic afferents suppress basolateral amygdala output through multiple, molecularly specific pathways ex vivo

We used an ex vivo preparation to determine unambiguously which types of BLA neurons are facilitated or suppressed by photostimulation of cholinergic basal forebrain neurons. We performed surgeries to fluorescently label two populations of BLA neurons to record their postsynaptic responses to photostimulation of basal forebrain cholinergic terminals (*Figure 7A*). Specifically, we investigated GABAergic neurons that are putative local interneurons and compared them to projector neurons (in this case those that project to mPFC), which are putative glutamatergic neurons. We used double-transgenic mice (ChAT::Cre × VGAT::flpo) to photostimulate basal forebrain cholinergic axonal terminals in the BLA through Cre-dependent expression of ChR2. We additionally fluorescently labeled BLA GABAergic neurons by flpo-dependent expression of eYFP. After at least 6 wk, we performed a second surgery to fluorescently label BLA neurons projecting to the medial prefrontal cortex (mPFC) through injection of a retrograde tracer (cholera toxin b subunit fused to Alexa-647, CTB-647) in the mPFC. One week after the second surgery, mice were euthanized and coronal slices of the BLA were prepared in order to perform whole-cell recordings from either fluorescently identified mPFC projecting BLA neurons (BLA-mPFC) or GABAergic VGAT::flpo BLA neurons (BLA$^{GABA}$), while photostimulating basal forebrain cholinergic axonal terminals (*Figure 7B and C*).

These two BLA neural populations were nonoverlapping and had strikingly different responses to photostimulation of cholinergic axonal terminals. BLA-mPFC neurons responded to cholinergic terminal photostimulation with a prolonged suppression (*Figure 7E and F*, *Figure 7—figure supplement 1*). In contrast, BLA$^{GABA}$ neurons responded to cholinergic photostimulation with a rapid and more transient facilitation. The amplitude of the fast excitatory postsynaptic potential (EPSP) was greater in BLA$^{GABA}$ neurons compared with BLA-mPFC neurons, while the inhibitory postsynaptic current (IPSP) amplitude was greater in BLA-mPFC neurons relative to BLA$^{GABA}$ (*Figure 7G*). These represented independent, direct monosynaptic responses to photostimulation of cholinergic afferents as each response persisted in the absence of spike-driven synaptic release (to eliminate indirect/polysynaptic transmission), blocked by tetrodotoxin (TTX) (*Figure 7H and I*).

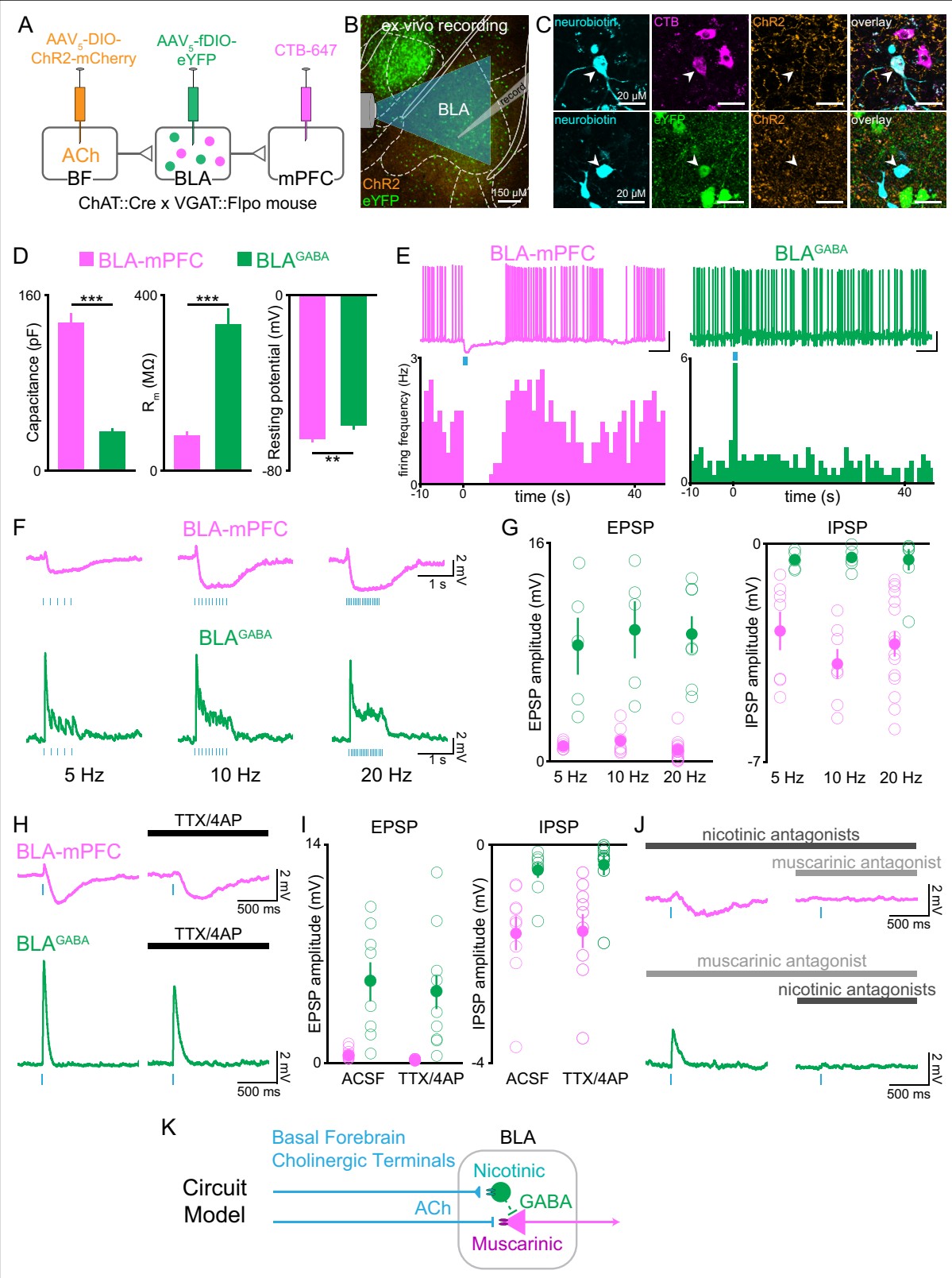

**Figure 7.** Cholinergic afferents suppress basolateral amygdala output through muscarinic receptors and feed-forward inhibition. (**A**) Schematic of injection strategy to express ChR2 in cholinergic neurons of the basal forebrain (BF) and eYFP in GABAergic neurons of the BLA (BLA^GABA) using conditional viral expression in ChAT::Cre × VGAT::Flpo mice (VGAT = vesicular GABAergic transporter), along with CTB-647 as a retrograde marker of neurons projecting to dorsomedial prefrontal cortex (dmPFC). (**B**) Confocal image of the basolateral amygdala (BLA) showing whole-cell patch-clamp

*Figure 7 continued on next page*

*Figure 7 continued*

recording arrangement in the BLA with optical stimulation of ChR2-expressing BF terminals. AP coordinate = –1.58. (**C**) High-magnification images of neurobiotin-filled recorded BLA neurons expressing CTB-647 (BLA-mPFC; upper panels) and eYFP (BLA$^{GABA}$, lower panels). (**D**) Passive membrane properties of BLA-mPFC and BLA$^{GABA}$ neurons. BLA-mPFC neurons had significantly greater capacitance (unpaired *t*-test: $t_{42}$ = 11.90, ***p<0.001, n = 20 BLA-mPFC, n = 24 BLA$^{GABA}$, from nine mice), smaller membrane resistance (unpaired *t*-test: $t_{42}$=6.326, ***p<0.001, n = 20 BLA-mPFC, n = 24 BLA$^{GABA}$, from nine mice), and more negative resting membrane potential (unpaired *t*-test: $t_{29}$ = 2.857, **p=0.0078, n = 13 BLA-mPFC, n = 18 BLA$^{GABA}$, from eight mice) than BLA$^{GABA}$ neurons. (**E**) Example trace and frequency histogram showing suppression of firing in BLA-mPFC neurons and facilitation of firing of BLA$^{GABA}$ neurons following optical stimulation of cholinergic terminals (470 nm light, 20 Hz; scale bars = 20 mV, 5 s). (**F**) Membrane potential of BLA-mPFC (upper traces) and BLA$^{GABA}$ neurons (lower traces) in response to 1 s 470 nm light delivered at 5, 10, and 20 Hz in current-clamp. (**G**) At each stimulation frequency, the amplitude of the fast excitatory postsynaptic potential (EPSP) was greater in BLA$^{GABA}$ neurons (green) compared with BLA-mPFC neurons (magenta; two-way ANOVA, main effect of cell type: $F_{1,40}$ = 95.59, ***p<0.001; n = 7, 7, and 15 BLA-mPFC neurons at 5, 10, and 20 Hz, n = 5, 5, and 7 BLA$^{GABA}$ neurons at 5, 10, and 20 Hz, from nine mice), while the slower inhibitory postsynaptic current (IPSP) was greater in BLA-mPFC neurons (two-way ANOVA, main effect of cell type: $F_{1,40}$ = 47.29, ***p<0.001; n = 7, 7, and 15 BLA-mPFC neurons at 5, 10, and 20 Hz, n = 5, 5, and 7 BLA$^{GABA}$ neurons at 5, 10, and 20 Hz, from nine mice). (**H**) Response of BLA-mPFC (upper traces) and BLA$^{GABA}$ neurons (lower traces) to a single 5 ms pulse of 470 nm light, with application of TTX/4AP to isolate monosynaptic currents. (**I**) Following application of TTX/4AP, the EPSP was maintained in BLA$^{GABA}$ neurons (green; unpaired *t*-test: $t_{15}$ = 0.367, p=0.719, n = 8 [ACSF] and n = 9 [TTX/4AP] BLA$^{GABA}$ cells from four mice), while the IPSP was maintained in BLA-mPFC neurons (magenta; unpaired *t*-test: $t_{16}$ = 0.094, p=0.926, n = 9 [ACSF] and n = 9 [TTX/4AP] BLA-mPFC cells from three mice). (**J**) Example traces showing inhibition of the IPSP in BLA-mPFC neurons (upper panels) by the muscarinic receptor antagonist scopolamine (10 μM) (dark gray), but not nicotinic antagonists (dihydro-ß-erythroidine 10 μM, methyllycaconitine 0.1 μM, mecamylamine 10 μM) (light gray), and inhibition of the EPSP in BLA$^{GABA}$ neurons (lower panels) by nicotinic receptor antagonists, but not muscarinic. (**K**) Proposed circuit model showing BF inhibition of BLA output by ACh acting at nicotinic receptors on BLA$^{GABA}$ neurons and muscarinic receptors on projection neurons. Dashed lines represent local BLA$^{GABA}$ neuron synapses onto BLA projector neurons from prior literature (*Lee and Kim, 2019*; *Woodruff and Sah, 2007*).

The online version of this article includes the following source data and figure supplement(s) for figure 7:

**Source data 1.** Passive membrane properties for the patched cells, as shown in *Figure 7D*.

**Source data 2.** The EPSP and IPSP amplitudes for patched cells, as shown in *Figure 7G, I*.

**Figure supplement 1.** Traces from individual neuron ex vivo slice recordings.

To understand how these neural populations could have such divergent responses to basal forebrain cholinergic inputs, we tested whether suppression in BLA-mPFC neurons and facilitation in BLA$^{GABA}$ neurons were regulated by different cholinergic receptor classes. Photostimulation-evoked IPSPs in BLA-mPFC neurons were blocked by a muscarinic antagonist, but not nicotinic antagonist (*Figure 7J*, top row). In contrast, the photostimulation-evoked facilitation in BLA$^{GABA}$ neurons was blocked by a nicotinic antagonist, but not a muscarinic antagonist (*Figure 7J*, bottom row). These results suggest a circuit model in which cholinergic afferents suppress BLA output through multiple, molecularly specific pathways, both through direct muscarinic suppression of projection neurons, as well as through nicotinic facilitation of GABAergic neurons, which can locally inhibit BLA projection neurons (*Lee and Kim, 2019*; *Woodruff and Sah, 2007*; *Figure 7K*). In total, the effects of cholinergic input to the BLA appear to be a suppression of BLA projector neuron output, and this effect may be most prominent when the system is primed to respond to cholinergic input by behavioral reinforcement.

## Discussion

### Basal forebrain cholinergic neurons can promote conditioned responding in the absence of discrete cues

Here we demonstrate that photostimulation of basal forebrain cholinergic neurons can promote conditioned responding, even in the absence of discrete external cues. ACh has long been posited to play an important role in directing attention to extrinsic stimuli, as measured by facilitating conditioned responses to such stimuli (*Parikh et al., 2007*; *Pinto et al., 2013*). Surprisingly, we observed that photostimulation of cholinergic basal forebrain neurons was sufficient to promote conditioned responding, even in the absence of other discrete stimuli (*Figure 1*). We observed direct behavioral (*Figure 1*) and neural responses (*Figure 6*), often used as readouts of attention, to photostimulation of basal forebrain cholinergic neurons.

## Reward availability modulates the impact of basal forebrain cholinergic neurons

Remarkably, photostimulation of basal forebrain cholinergic neurons only drove conditioned responding when paired with the opportunity to collect rewards (*Figure 1*). Basal forebrain neurons, including cholinergic neurons, are active at the time of both positively and negatively valenced reinforcers (*Hangya et al., 2015*; *Harrison et al., 2016*; *Peck and Salzman, 2014*). Our results demonstrate that at least some part of the previously described reward associations may be due to reward collection behavior such as licking, rather than only reward per se. Previous work has also suggested that ACh may amplify the effects of reinforcers during learning and strengthen plasticity (*Jiang et al., 2016*). In the current study, reward availability modified the behavioral and neural effects of basal forebrain cholinergic neuron photostimulation itself. BLA neurons became more suppressed as a population (*Figure 6*), appearing to become more similar to findings in an ex vivo preparation from behaviorally naïve animals (*Figure 7*). Taken together, these data suggest a model wherein effects of ACh may be unmasked or amplified by reward availability.

## The effects of basal forebrain cholinergic neurons may be dynamically gated postsynaptically

Given the modulation of ACh effects by reward availability, we investigated whether this difference was due to a difference in the amount of ACh elicited by photostimulation. Cholinergic terminals can express presynaptic receptors, including cholinergic receptors, which may modulate ACh release (*Muller et al., 2016*; *Thany and Tricoire-Leignel, 2011*). If photostimulation induced a different amount of ACh release during Photostim-Reinforced sessions, this may have explained the change in conditioned behavior. However, photostimulation of cholinergic terminals in the BLA evoked similar levels of ACh in Photostim-Unreinforced and Photostim-Reinforced sessions (*Figure 5*). This suggests that the similar levels of ACh instead may have been gated or differentially interpreted by downstream, postsynaptic neurons, depending on reward availability. The postulation of such gating merits future investigation, but may be mediated by coincident signals of reinforcement to the BLA, such as dopaminergic inputs (*Lutas et al., 2022*; *Lutas et al., 2019*; *Tye et al., 2010*), given that dopaminergic receptors are expressed by BLA neurons that also express cholinergic receptors (*Równiak et al., 2017*). Although dmPFC neurons were responsive to photostimulation of basal forebrain cholinergic neurons, these responses did not appear to be dependent on the association between photostimulation with reward, despite the robust gating effects of dopamine in dmPFC (*Vander Weele et al., 2018*).

## Basal forebrain cholinergic neurons reflect a transition from conditioned stimulus to response

Rather than changing how other stimuli are processed from neutral into conditioned stimuli, our results suggest that cholinergic activity can itself become conditioned. The lack of response to photostimulation during Photostim-Unreinforced sessions provides important constraints on possible interpretations (*Figure 1*). Mice were in a familiar context wherein the only meaningful behavioral response was to lick for unpredictable rewards. However, photostimulation of basal forebrain cholinergic neurons during Photostim-Unreinforced sessions did not promote conditioned responding in these sessions, as might be predicted by a number of alternative interpretations, such as nonspecific increases in movement, arousal, or contextual awareness.

Our findings suggest that photostimulated release of ACh can have a similar function as conditioned stimuli or cues, able to trigger conditioned responses. Recent work has shown that basal forebrain cholinergic neurons are active following presentation of conditioned stimuli (*Crouse et al., 2020*; *Guo et al., 2019*; *Sturgill et al., 2020*). Additionally, work in visual cortex has suggested that ACh may help link stimuli with the time of expected rewards (*Chubykin et al., 2013*; *Liu et al., 2015*). We also demonstrated, however, that cholinergic neurons were consistently active at the times of conditioned responding, even in the absence of cues and rewards. This suggests that cholinergic transients are poised to play a role in conditioned responding and serve to signal more than just a salient event promoting a response. Indeed, blocking cholinergic muscarinic receptors impaired the ability of mice to respond even to conditioned tones (*Figure 2*). This is consistent with other studies in which

inhibition of cholinergic terminals in the BLA prevented mice from expressing freezing behavior at the time of fear conditioning (*Jiang et al., 2016*).

Our observation on the impact of ACh on conditioned behavior also provides new context for interpreting several prior results that have examined the role of basal forebrain cholinergic neurons in attention to external stimuli. Interestingly, even when photostimulation of cholinergic basal forebrain neurons has previously been noted to increase discriminability between stimuli, it appears to do this primarily by increasing conditioned responses to conditioned stimuli, rather than suppressing false alarms (*Pinto et al., 2013*). Additionally, in sustained attention tasks to report the presence of stimuli to collect rewards, photostimulation of basal forebrain cholinergic neurons promotes conditioned responding as if cues are present, even in their absence (*Gritton et al., 2016*), and immunotoxic lesions of basal forebrain cholinergic neurons lead to omissions of any response at all (*McGaughy et al., 2002*).

The responses of cholinergic basal forebrain neurons (*Figure 3*) and ACh release within the BLA (*Figures 4 and 5*) indicate that conditioned responses of licking are represented by cholinergic signaling even in the absence of conditioned stimuli or reward delivery, reminiscent of signals seen in lateral hypothalamic (LH) neurons projecting to the ventral tegmental area (VTA) (*Nieh et al., 2015*). While LH-VTA neurons were capable of promoting compulsive sucrose seeking behavior, we do not observe the same stereotyped motor sequences in cholinergic neurons as when disinhibiting VTA dopamine neurons via the LH-VTA pathway (*Nieh et al., 2016*), and the ability of cholinergic signaling to promote compulsivity begs further exploration.

## Comparisons to other studies of photostimulation of cholinergic basal forebrain neurons

Cholinergic neurons from the basal forebrain have different projections depend on their location in the basal forebrain (*Zaborszky and Gyengesi, 2012*), and even within regions may be physiologically heterogeneous (*Laszlovszky et al., 2020*). While photostimulation of anterior basal forebrain cholinergic neurons projecting to the lateral hypothalamus suppresses appetite (*Herman et al., 2016*), here, photostimulation of the more posterior cholinergic population in the sublenticular substantia innominata/extended amygdala increased consummatory behavior (*Figure 1*). We did not observe an effect on locomotion, in both an unrewarded and a rewarded context, despite the physiological correlation of cholinergic neuron activity with locomotion (*Harrison et al., 2016*).

Additionally, while other work has suggested that photostimulation of basal forebrain cholinergic axonal terminals in the BLA can be rewarding (*Aitta-Aho et al., 2018*), we did not observe a reinforcing effect of somatic basal forebrain cholinergic photostimulation. In addition to differences in the site of photostimulation, we transfected a more posterior and lateral portion of the basal forebrain (+0.05 mm AP in Aitta-Aho et al. vs –0.4 mm AP here, and ± 1.15 mm ML in Aitta-Aho et al. vs ± 1.80 mm here). This more anterior portion of the basal forebrain, closer to the horizontal limb of the diagonal band of Broca (HDB), is more responsive to reward omission, more accurately classifies behavioral outcomes, and more closely tracks fluctuations in pupil-indexed global brain state (*Robert et al., 2021*). In contrast, cholinergic neurons in the more posterior portions of the basal forebrain are more responsive to unconditioned auditory stimuli, orofacial movements, aversive reinforcement, and showed robust associative plasticity for punishment-predicting cues (*Robert et al., 2021*).

While it is possible that some of the effects of photostimulation of cholinergic terminals within the BLA were mediated by backfiring of cholinergic axons, several lines of evidence suggest that basal forebrain-BLA cholinergic projection is a relatively distinct population from basal forebrain cholinergic neurons projecting more widely to cortex: (1) basal forebrain cholinergic neurons that project to the BLA develop earlier embryologically than those that project to cortex and hippocampus (*Allaway et al., 2020*). (2) Basal forebrain cholinergic neurons that project to the BLA rarely have collaterals (*Carlsen et al., 1985*). (3) Basal forebrain cholinergic neurons that project to the BLA are sensitive to different immunotoxins and neurotoxins than cholinergic neurons that project to cortex and hippocampus (*Beninger et al., 2001*; *Boegman et al., 1992*; *Heckers et al., 1994*; *Hecker and Mesulam, 1994*). This suggests that backfiring from cholinergic axons in the BLA to the soma and then to cortex is likely to be limited.

## Cell type-specific responses to ACh

We noted heterogeneous responses among BLA neurons in vivo to photostimulation of basal forebrain cholinergic neurons (*Figure 6*). Heterogeneous responses have also been previously noted ex vivo (*Unal et al., 2015*; *Washburn and Moises, 1992*). Through a double transgenic approach, we determined that these heterogeneous responses could be explained on a cellular level, with projecting, putative glutamatergic neurons suppressed, and GABAergic, putative local interneurons neurons facilitated by cholinergic inputs (*Figure 7*). The BLA GABAergic neuron population is thought to represent local interneurons that inhibit projection neurons (*Washburn and Moises, 1992*). Although both excitation of GABAergic neurons and inhibition of mPFC projection neurons were monosynaptic, these cholinergic effects could in concert function to suppress BLA output (*Lee and Kim, 2019*; *Pidoplichko et al., 2013*; *Zhu et al., 2005*), which may decouple regions downstream from the BLA, such as the dmPFC, and allow them to function independently with regard to cholinergic input or other inputs.

This cellular specificity was mediated by different receptors, with short-latency facilitation in GABAergic neurons mediated by nicotinic receptors and longer-latency suppression in projection neurons mediated by muscarinic receptors (*Figure 7*). Prior histological work had suggested that M1 muscarinic receptors are predominantly expressed by pyramidal neurons in the BLA (*McDonald and Mascagni, 2010*), and interneurons express nicotinic receptors (*Pidoplichko et al., 2013*). Functional differences similar to those we have observed between excitatory and inhibitory neurons have been seen in somatosensory cortex (*Dasgupta et al., 2018*). Strikingly, the behavioral response to even brief photostimulation of basal forebrain cholinergic neurons occurred by approximately 0.5 s (*Figure 1*). Surprisingly, this sub-second response appeared primarily mediated by slower muscarinic receptors rather than faster nicotinic receptors (*Figure 2*). It is possible that these mechanisms might differ based on additional cholinergic factors, such as local interneurons in sensory cortex as well as behavioral context such as reward vs. shock conditioning (*Letzkus et al., 2011*). Our pharmacology experiments suggest that the effects of basal forebrain cholinergic neurons on nicotinic receptors are insufficient to trigger conditioned responding, although it is possible that we did not block all types of nicotinic receptors. Projection neurons, which receive a preponderance of cholinergic input (*McDonald et al., 2011*), may serve as a critical point of convergence for ACh influence over BLA networks.

## The effects of augmenting ACh levels may depend on context

Multiple mental health disorders involve cholinergic deficiency, including delirium, schizophrenia, and attention-deficit hyperactivity disorder (*Higley and Picciotto, 2014*; *Hshieh et al., 2008*; *Potter et al., 2014*). Each of these disorders is associated with severe cognitive and functional impairments (*American Psychiatric Association, 2013*), symptoms that impact not only patients (*Morandi et al., 2015a*), but also their families and caregivers (*Morandi et al., 2015b*; *O'Malley et al., 2008*). Disappointingly, current strategies to augment cholinergic function, such as cholinesterase inhibitors, have not had a strong clinical impact on several neuropsychiatric illnesses such as dementia, delirium, schizophrenia, and ADHD because of both insufficient benefit and significant side effects (*Biederman et al., 2006*; *Cubo et al., 2008*; *Siddiqi et al., 2016*; *Singh et al., 2012*). While there are many reasons that cholinergic augmentation is unlikely to treat any of these complex diseases completely, our translationally motivated concern is that augmenting cholinergic levels, even with temporal and spatial specificity here through optogenetics, may have unpredictable effects depending upon the reward context surrounding this augmentation. Additionally, it is possible that extrinsic augmentation of cholinergic tone may be processed differently than intrinsic fluctuations in cholinergic tone. It is possible that coupling cholinergic augmentation with other therapies, such as cognitive or behavioral therapies, that can explicitly incorporate rewards or proxies for reinforcers, may provide a new opportunity for more sustained and predictable benefits to patients.

# Materials and methods

**Key resources table**

| Reagent type (species) or resource | Designation | Source or reference | Identifiers | Additional information |
|---|---|---|---|---|
| Strain, strain background (*Mus musculus*) | B6N.ChAT-IRES-Cre | Jackson Laboratory | IMSR_JAX:018957 | |
| Strain, strain background (*M. musculus*) | B6.Cg-Slc32a1$^{tm1.1(flpo)Hze}$/J | Jackson Laboratory | IMSR_JAX:029591 | |
| Strain, strain background (*M. musculus*) | C57BL/6NJ | Jackson Laboratory | IMSR_JAX:005304 | |
| Strain, strain background (*AAV*) | AAV5-EF1a-DIO-hChR2(H134R)-eYFP | UNC Vector Core | AV4313X | |
| Strain, strain background (*AAV*) | AAV5-EF1a-DIO-eYFP | UNC Vector Core | AV4310G | |
| Strain, strain background (*AAV*) | AAVDJ-EF1a-DIO-GCaMP6s | Stanford Vector Core | 2612 | |
| Strain, strain background (*AAV*) | AAV5-Syn-FLEX-rc [ChrimsonR-tdTomato] | Addgene | 62723-AAV5 | |
| Strain, strain background (*AAV*) | AAV1-CAG-FLEX-tdTomato | UNC Vector Core | AAV-CAG-FLEX-tdTomato | |
| Strain, strain background (*AAV*) | AAV9-hSyn-Ach4.3 (h-A06) | ViGene | h-A06 | |
| Chemical compound | Tetrodotoxin | Tocris | 4368-28-9 | |
| Chemical compound | 4-Aminopyridine | Sigma-Aldrich | 504-24-5 | |
| Chemical compound | Scopolamine | Sigma-Aldrich | S0929-1G | |
| Chemical compound | Mecamylamine hydrochloride | Sigma-Aldrich | M9020-5MG | |
| Chemical compound | Methyllycaconitine citrate salt | Sigma-Aldrich | M168-5MG | |
| Chemical compound | Dihydro-ß-erythroidine | Tocris | 2349 | |
| Chemical compound | VECTASHIELD HardSet Antifade Mounting Medium with DAPI | VectorLabs | H-1500 | Mounting medium |
| Antibody | Anti-goat-Alexa Fluor 647 (donkey polyclonal) | Jackson ImmunoResearch | 705-605-147 | 1:500 |
| Antibody | Anti-choline acetyltransferase (goat polyclonal) | EMD Millipore | AB144P-1ML | 1:100/1:200 |
| Chemical compound | Buprenorphine hydrochloride | MIT Veterinary Pharmacy | NDC 12496-0757-5 | Analgesic |
| Software | MATLAB | MathWorks | https://www.mathworks.com/ | |
| Software | R | R Foundation for Statistical Computing | http://www.R-project.org/ | |
| Software | Adobe Illustrator | Adobe | https://www.adobe.com/products/illustrator.html | |

*Continued on next page*

*Continued*

| Reagent type (species) or resource | Designation | Source or reference | Identifiers | Additional information |
|---|---|---|---|---|
| Software | Offline Sorter | Plexon | https://plexon.com/products/offline-sorter/ | |
| Software | Ethovision XT | Noldus | https://www.noldus.com/ethovision-xt | |
| Software | pClamp 10.4 software | Molecular Devices | https://www.moleculardevices.com/products/axon-patch-clamp-system/acquisition-and-analysis-software/pclamp-software-suite | |
| Other | Optic fiber | Thorlabs | TS1843490 | See 'General stereotaxic surgery methods' |
| Other | Ferrules | Kientec Systems | FSS-LC-330 | See OFT, RTPP tests in 'Materials and methods' |
| Other | Small animal stereotaxic frame | David Kopf Instruments | Model 942 | See 'General stereotaxic surgery methods' |
| Other | USB cameras | ELP | ELP-USBFHD01M-RL36 | See ' Pupillometry' |
| Other | 0.10 mL Microsyringe | World Precision Instruments | NANOFIL-NF33BL-2 | See 'General stereotaxic surgery methods' |
| Other | Microsyringe Pump UMP3 and Controller Micro4 | World Precision Instruments | UMP3-3 | See 'General stereotaxic surgery methods' |
| Other | Peristaltic pump for ex vivo recordings | Minipuls 3 Gilson | F155001 | See 'Ex vivo electrophysiology recordings' |
| Other | Miniature Inert Liquid Solenoid Valve | Parker Hannifin | 003-0218-900 | See ' Behavioral system control' |
| Other | ATmega328 Arduino | Digi-Key Electronics | 1050-1001-ND | See 'Head-fixed behavioral and optogenetic equipment' |
| Other | Speaker | Digi-Key Electronics | GF0401M-ND | See 'Cued tone task methods' |
| Other | Laser Shutter Heads | Stanford Research Systems | SR475 | See 'Optogenetic photostimulation of basal forebrain cholinergic neurons during head-fixed behavior' |
| Other | Rotary Encoder | US Digital | E2-200-250-NE-D-D-B | See 'Treadmill' |
| Other | Multifunction I/O Device | National Instruments | USB-6211 | See 'General stereotaxic surgery methods' |
| Other | Pulse stimulator | A.M.P.I. | Master-8 | See 'Optogenetic photostimulation of basal forebrain cholinergic neurons during head-fixed behavior' |
| Other | 5 mm White LED | Digi-Key Electronics | C513A-WSN-CY0Z0341-ND | See 'Head-fixed behavioral and optogenetic equipment' |
| Other | Electrodes: 22.9 µm nichrome wire | California Fine Wire | Stablohm 675 | See 'In vivo electrophysiology recordings' |
| Other | Gold Non-Cyanide | Sifco asc | 80535500 | See 'In vivo electrophysiology recordings' |
| Other | Open Ephys Acquisition Board | Open Ephys | C3324 | See 'In vivo electrophysiology recordings' |
| Other | HM430 Microtome | Thermo Fisher Scientific | 910010 | See 'In vivo electrophysiology histology,' 'Ex vivo electrophysiology histology,' 'Immunohistochemistry' |

*Continued on next page*

*Continued*

| Reagent type (species) or resource | Designation | Source or reference | Identifiers | Additional information |
|---|---|---|---|---|
| Other | Confocal Laser-Scanning microscope | Olympus | FV1000 | See 'In vivo electrophysiology histology,' 'Ex vivo electrophysiology histology,' 'Immunohistochemistry' |
| Other | Diode Blue 473 nm Laser | OptoEngine LLC | MBL-III-473/1-100mW | See 'Optogenetic photostimulation of basal forebrain cholinergic neurons during head-fixed behavior, OFT, RTPP' |
| Other | Horizontal puller for glass microelectrodes for ex vivo recordings | Sutter | P-1000 | See 'Ex vivo electrophysiology recordings' |
| Other | Multiclamp amplifier for ex vivo recordings | Molecular Devices | 700B | See 'Ex vivo electrophysiology recordings' |
| Other | Microscope for ex vivo recordings | Olympus | BX51 | See 'Ex vivo electrophysiology recordings' |

## Subjects

Female and male hemizygous ChAT::Cre mice (*Chen et al., 2018*; *Rossi et al., 2011*) were group housed by sex until surgery on a reversed 12 hr light-dark cycle in a humidity and temperature-controlled vivarium. All behavioral experiments were conducted during the dark phase of the animals' cycle. All experiments involving the use of animals were in accordance with National Institutes of Health guidelines and approved by the Massachusetts Institute of Technology's Committee on Animal Care.

## General stereotaxic surgery methods

General surgical methods are provided here and specific subject/surgery details for each experiment are detailed in their respective sections below. Surgeries were performed prior to behavioral training and all other experiments. For all mice, surgeries were performed under aseptic conditions and body temperature was maintained with a heating pad. Mice were anesthetized with isoflurane in oxygen (4% for induction, 1–2% for maintenance, 0.8 L/min oxygen flow rate). Following induction, we shaved the scalp and placed the subjects on a digital small animal stereotaxic instrument (David Kopf Instruments). Ophthalmic ointment was applied to the eyes and the incision area was scrubbed three times with alternating betadine and 70% ethanol. An incision was made along the midline to expose the skull, which was then leveled. All measurements for virus injections and implants were made relative to Bregma using the approximated intersection of skull sutures. A dental drill was used to perform small craniotomies (EXL-M40, Osada).

Viral injections were performed using a beveled 33-gauge microinjection needle connected to a 10 µL microsyringe (Nanofil; WPI, Sarasota, FL) at a rate of 100 nL/min using a microsyringe pump (UMP3; WPI) and pump controller (Micro4; WPI). After injections were complete, 10 min were allowed to pass before the needle was slowly withdrawn. Optic fibers and/or electrodes were then implanted. For head-fixation, a 2mm × 2mm × 25 mm aluminum headbar was placed horizontally over Lambda. A layer of adhesive cement (C&B Metabond; Parkell Inc, NY) was used to secure the implants and headbar to the skull, followed by a black cranioplastic cement (Ortho-Jet; Lang, IL) to prevent light escape. The cement was allowed to dry completely before closure of the incision with 4.0 nylon sutures.

Subjects received a perioperative subcutaneous injection of sustained release buprenorphine (1 mg/kg) for analgesia. During recovery, subjects were also injected subcutaneously with 1 ml of warm Lactated Ringers solution and kept on a heat pad until fully recovered from anesthesia. For all experiments involving viral or tracer injections, animals containing mistargeted injections were excluded after histological verification.

## Stereotaxic surgery for optogenetic photostimulation of basal forebrain cholinergic neurons

For optogenetic photostimulation of basal forebrain cholinergic neurons, 500 nL of an adeno-associated virus encoding either channelrhodopsin (AAV5/EF1a-DIO-hChR2(H134R)-eYFP; UNC, 5.5

× 10$^{12}$) or a control fluorophore (AAV5/EF1a-DIO-eYFP; UNC, 4.4 × 10$^{12}$) were injected bilaterally into basal forebrain of each hemisphere at AP –0.4, ML ± 1.8, DV –4.7. For photostimulation of cholinergic neurons at their basal forebrain soma, optic fibers (300 µm diameter) were implanted bilaterally into the basal forebrain at AP –0.4, ML ± 1.8, DV –4.3. For photostimulation of cholinergic neuron axon terminals in the BLA, optic fibers (300 µm diameter) were implanted unilaterally above the BLA at AP –1.4, ML ± 3.1, DV –4.6.

## Head-fixed behavioral and optogenetic equipment

Head-fixed boxes were custom built using various optomechanical components (Thorlabs) and 3D-printed parts mounted on top of a solid aluminum optical breadboard (SAB0810, Base Optics) housed within a 19 quart drybox (UC19-YHV, Engel Coolers). Mice were head-fixed on either a 3D-printed rectangular platform or a custom 3D-printed linear-belt treadmill, in front of a blunt 18G needle spout (75165A754, McMaster Carr). For electrophysiology experiments, licks were registered using an infrared beam passing in front of the spout tip (emitter 935 nm, OP165A, TT Electronics/ Optek Technology; phototransistor SFH 309 FA-4/5, OSRAM Opto Semiconductors). For other experiments, licks were registered using a capacitive contact circuit (MPR121, Adafruit) that was interpreted by a microcontroller (Arduino Uno SMD R3 ATMEGA328, Arduino). A white LED module (1621, Adafruit) provided a low level of ambient light.

## Behavioral system control

All behavior for each box was controlled by a microcontroller (Arduino Mega 2560 Rev3, Arduino), which ran customized behavioral code to register licks, deliver fluid, present tones, and trigger photostimulation. Fluid rewards consisted of 4 µL of a sweet caloric fluid (Vanilla Ensure Original Shake, Abbott). Fluid delivery was controlled by opening a solenoid valve (003-0860-900, Parker, NH) and delivered to the spout by gravity flow via plastic acrylic tubing (McMaster-Carr, IL). All valves were calibrated using timing duration to ensure consistent fluid volumes.

## Head-fixed behavioral training

After at least 3 wk of recovery from surgery, animals were food restricted and maintained on at least 85% body weight. Animals had free access to water. After stable food restriction, training was initiated. On each day, mice received enough food supplementation using standard lab chow to maintain their body weight between 85 and 90% of their free body weight (typically 2.5–5 g).

## Handling

Mice were first handled for 5 min a day for 5 d to reduce stress and increase familiarity with the experimenter. During this time, they also were given fluid rewards via a hand held plastic pipette.

## Uncued head-fixed training

For uncued head-fixed training, mice were head-fixed in the behavioral box and a lickspout was placed close to the mouth. The animal received a few drops of fluid reward to initiate licking. Once the mouse started licking, the spout was retracted away slightly to a distance still reachable by licking.

Sessions were divided into trials of 3 s windows with intervening unrewarded ITIs. During early training, 90% of the 3 s windows were designated as *Windows of Opportunity*. If mice licked during the *Windows of Opportunity*, they received a 4 µL fluid reward after a brief delay (0.1 s). Since rewarded windows were not cued, mice did not know when they initiated a lick whether it would be rewarded, making the rewards unpredictable. Only the first lick within a window resulted in reward delivery. The ITIs between *Windows of Opportunity* were randomly selected from an exponential distribution between 3 and 6 s (mean 4 s). The remaining 10% of 3 s windows were designated as matched *Unrewarded Windows*. If mice licked during this time, no reward was delivered, similar to the rest of the unrewarded ITIs.

The first session was 30 min long and subsequent sessions were 1 hr long. Mice were trained daily until they attempted to collect rewards on at least 30% of uncued *Windows of Opportunity* in order to avoid ceiling and floor effects of subsequent manipulations. The likelihood of licking was defined as percent of 3 s windows, for either Rewarded or *Unrewarded Windows*, in which mice licked at least once. Mice underwent approximately 7 d of uncued head-fixed training to reach criterion. Single

measurements were compared between ChR2 and eYFP mice using rank-sum tests in R (*R Development Core Team, 2015*). Repeated measurements from the same mice were analyzed using linear mixed effects models fit by restricted maximum likelihood using the lme4 package (*Bates et al., 2015*). Fixed effects included virus group (ChR2 vs. eYFP) and window type (Rewarded vs. Unrewarded), as well as their interaction. Random effects were modeled using random intercepts for each subject. p-values were obtained using the lmerTest package (*Kuznetsova et al., 2017*).

## Optogenetic photostimulation of basal forebrain cholinergic neurons during head-fixed behavior

Following initial uncued head-fixed training, ChR2 and eYFP mice were tested to examine whether photostimulation of basal forebrain cholinergic neurons affected licking, the conditioned response. For optogenetic photostimulation, a 473 nm diode laser (MBL-III-473/1–100 mW, OptoEngine LLC) was used as a light source. Photostimulation was delivered at 4 mW (measured at the fiber tip) using 5 ms pulses delivered at 20 Hz. Lasers were continuously on to minimize power fluctuations, but photostimulation was gated at the source prior to entering a collimator (HPUC-23AF-473-S-11AS-LBH-BL-SP, OZ Optics) through a laser shutter head (SR475, Stanford Research Systems), controlled by a four-channel laser shutter driver (SR474, Stanford Research Systems). Laser light was then routed from the collimator through patch cords (Doric, Québec, Canada) and split for bilateral photostimulation using a rotary joint (1 × 2 Fiber-optic Rotary Joints – Intensity Division; Doric, Québec, Canada), with a subsequent patch cord terminating on the implanted ferrule. All connections including that to the implanted ferrule were optically shielded to prevent light leakage.

### Photostim-unreinforced sessions

To assess innate behavioral responses to photostimulation of basal forebrain cholinergic neurons, mice underwent one session in which they received photostimulation of cholinergic basal forebrain neurons during unrewarded ITIs (Photostim-Unreinforced sessions). On a subset of ITIs, approximately once a minute, photostimulation was delivered for 2 s, and the likelihood and latency of licking was recorded in the 3 s following photostimulation onset to assess effects on conditioned responding.

During this session, if mice licked during unsignaled *Windows of Opportunity*, which did not have photostimulation, they continued to receive rewards. ITIs between *Windows of Opportunity* were randomly selected from an exponential distribution between 4.5 and 10 s (mean 5 s). Licking during ITIs had no consequence, which included photostimulation during these sessions. Photostimulation was delivered at least 3.5 s away from either the beginning or end of a *Window of Opportunity*.

### Photostim-reinforced sessions

During Photostim-Reinforced sessions, 2 s of photostimulation was now delivered starting at the onset of the 3 s *Windows of Opportunity*. Photostimulation was delivered only on a subset of *Windows of Opportunity* (15%). Approximately one photostimulation trial was delivered per minute, thereby making up the minority of total session time and a minority of *Windows of Opportunity*. Licking during *Windows of Opportunity* yielded a fluid reward, whether the mouse received photostimulation or not.

Mice ran on two Photostim-Reinforced sessions, and behavior was analyzed from the second session, based on a priori planning in order to minimize multiple post hoc comparisons. Data from the planned analysis is shown in *Figure 1*, with data from all sessions in *Figure 1—figure supplement 2*. We analyzed data using linear mixed effects models, given the repeated measures from mice. Fixed effects included Virus (ChR2 vs. eYFP), Photostimulation trial types (Photostim vs. No photostim), and Reinforcement Session type (Photostim-Reinforced vs. Photostim-Unreinforced), as well as first- and second-order interactions. Random effects were modeled using random intercepts for each subject. Post hoc tests for all linear mixed effects models were performed using the emmeans package (*Wieduwilt et al., 2020*), with the Kenward–Roger method for degrees of freedom, and the Sidak method for p-value adjustment for multiple comparisons.

During a subsequent session, we varied photostimulation parameters to assess how little or brief the stimulation needed to be to produce conditioned responding. We randomly delivered photostimulation on *Windows of Opportunity* using either 1, 2, or 10 pulses of light at 20 Hz, corresponding to up to 0.5 s of photostimulation. These continued to be delivered at the onset of 3 s *Windows of Opportunity*. Fixed effects for linear mixed effects model analysis included Virus (ChR2 vs. eYFP),

photostimulation pulses (0, 1, 2, 10), and their interaction. Random effects were modeled using random intercepts for each subject. Post hoc tests corrected for multiple comparisons included comparisons between ChR2 and eYFP at each number of pulses, and comparisons within each virus group for 1, 2, or 10 pulses to 0 pulses.

## Treadmill

To assess the effects of basal forebrain photostimulation on locomotion, we collected locomotion data from a custom-made linear belt treadmill in the head-fix setup. Locomotion data was collected using a rotary encoder (US Digital E2-200-250-NE-D-D-B), digitized using a NI-USB-6211 at 1 kHz, and analyzed using MATLAB. There was no behavioral consequence of running. Behavioral event timings, including trial onset, photostimulation, licking, and reward delivery, were synchronized by sending TTLs from the Arduino microcontroller into an R-2R resistor ladder to multiplex the events into a single analog input channel on the NI-USB-6211.

Treadmill locomotion was analyzed around photostimulation, starting 2 s before photostimulation onset until 4 s after. Photostimulation duration was 2 s. Data was binned in 0.1 s bins for peri-event time histograms. Each bin was compared between ChR2 and eYFP mice for each session type (Photostim-Unreinforced or Photostim-Reinforced). Statistical significance of each bin was tested between groups using rank-sum tests, p<0.01. To compare photostimulation-evoked locomotion between sessions for the same mice and between groups, we used linear mixed effects models to analyze the mean locomotion during photostimulation (0–2 s). Fixed effects included Virus (ChR2 vs. eYFP), Reinforcement Session type (Photostim-Reinforced vs. Photostim-Unreinforced), as well as their interaction. Random effects were modeled using random intercepts for each subject.

## Pupillometry

In order to assess changes in arousal, we recorded pupil diameter during behavioral sessions. An infrared USB camera (ELP-USBFHD01M-RL36, ELP) was placed 15 cm from the mouse's eye. The infrared LED/emitter panel was unscrewed and pointed away from the mouse to decrease eye secretions. Ambient light was adjusted to keep the pupil size at an intermediate level, which allowed the pupil to fluctuate over a dynamic range. Recordings were started and stopped at the same time as behavior using Processing 2.2.1 running on a Windows desktop computer (Hewlett-Packard), which also initiated the Arduino microcontroller running behavior. Files were saved at 30 Hz as 640 × 480 pixel ogg vorbis video files. Behavioral event timings were identified for subsequent synchronization using 940 nm infrared LEDs (IR204, Everlight Electronics, Digi-Key) to signal behavioral events, including trial onset, photostimulation, licking, and reward delivery.

For pupil and body part tracking we used DeepLabCut (version 2.0.8) (*Mathis et al., 2018*; *Nath et al., 2019*). We labeled eight points for the right pupil of each mouse, according to cardinal and intercardinal compass directions (North, NorthEast, East, SouthEast, South, SouthWest, West, and NorthWest). Specifically, we labeled 483 frames taken from 25 videos from 21 animals, then 95% were used for training. We used a ResNet-50-based neural network with default parameters for 1.03 million training iterations (*He et al., 2016*; *Insafutdinov et al., 2016*). We validated with a single shuffle and found the test error was 1.08 pixels, train: 1.93 pixels. We then used a p-cutoff of 0.9 to condition the X,Y coordinates for future analysis. This network was then used to analyze videos from similar experimental settings. Relative pupil diameters were determined by calculating the distance for each major axis (North-South, East-West, NorthWest-SouthEast, NorthEast-SouthWest), and then taking the mean of these four measurements.

Pupil diameter was normalized to the diameter prior to photostimulation (−2 to −1 s relative to photostimulation onset), and the pupil diameter from each frame for each trial was expressed as the percent difference from baseline. Relative pupil diameter changes at each frame were compared between ChR2 and eYFP mice using rank-sum testing (p<0.05).

To compare photostimulation-evoked pupil diameter changes between sessions for the same mice and between groups, we used linear mixed effects models to analyze the mean pupil diameter following events onsets (0–4 s), separately for photostimulation and for reward delivery. Fixed effects included Virus (ChR2 vs. eYFP), Reinforcement Session type (Photostim-Reinforced vs. Photostim-Unreinforced), as well as their interaction. Random effects were modeled using random intercepts for

each subject. Post hoc tests corrected for multiple comparisons included four comparisons between ChR2 and eYFP for each session, and between sessions for each viral group.

## Cued tone task

In a separate set of sessions, mice were trained to respond to 3.5 kHz or 12 kHz tones (frequencies counter balanced between mice). Tones were generated using the tone function on the Arduino Mega, passed through analog low- and high-pass filters, and presented using an 8 ohm speaker (GF0401M, CUI Devices) at 55–60 dBA. Tones were presented for 2 s, and mice were rewarded for licking during the 3 s response window. These 1 hr sessions included only tone trials to signal rewards opportunities – there were no longer any unsignaled *Windows of Opportunity*.

## Interleaved tone or photostim-reinforced trials session

After 3 d of training on the cued tone task, mice were tested on a session which included interleaved tone or Photostim-Reinforced windows. Then, 2 s of either tone or photostimulation were presented, and mice could receive a reward if they licked within 3 s of the onsets. Only 90% of lick responses on each trial type were rewarded. Tone and photostimulation trials were equally likely, and were separated by ITIs with a mean of 12 sec (range 7–22 s, exponential distribution). Baseline/ITI licking was assessed by a priori statistically identifying 3 s windows between tone and photostimulation trials, with similar ITIs.

## In vivo cholinergic antagonist pharmacology

In order to test which class of receptors mediated the effects of photostimulation of cholinergic basal forebrain neurons, mice were injected with cholinergic receptor antagonists prior to an Interleaved Tone or Photostim-Reinforced Trials Session. Using a 27-gauge needle, we injected mice with either the muscarinic receptor antagonist scopolamine (0.3 or 1 mg/kg) (*Chintoh et al., 2003*) or the nicotinic receptor antagonist mecamylamine (1 mg/kg) (*Adermark et al., 2014*; *Zachariou et al., 2001*). These drugs and doses were chosen in discussion with an expert in the field (Dr. Marina Picciotto, personal communications). Injections were made 10 min before behavioral sessions. All drugs were dissolved in sterile saline and injected at 10 mL/kg volumes. Sterile saline alone was used for control injections. Drug injection orders were counterbalanced across animals.

We analyzed data using linear mixed effects models, given the repeated measures from mice. Fixed effects included virus (ChR2 vs. eYFP), trial types (tone, photostim, or ITI), and drug session type (saline, scopolamine 0.3 mg/kg, scopolamine 1 mg/kg, mecamylamine 1 mg/kg), as well as first- and second-order interactions. Random effects were modeled using random intercepts for each subject. Post hoc tests corrected for multiple comparisons were stratified by sessions, and included comparisons between ChR2 and eYFP for each trial type, and comparisons within each virus group for all three pairs of trials types.

## Open-field test

To assess whether BF cholinergic stimulation had an effect on locomotion, animals underwent an open-field test (*Matthews et al., 2016*). We attached fiber optic patch cables to the implanted ferrules on mice. Mice were then placed in the center of an open 50 × 53 cm arena composed of four transparent Plexiglas walls illuminated by 30 lux ambient light. They were allowed to freely move throughout the arena for 15 min. A video camera was positioned directly above the arena to track the movement of each mouse throughout the session (EthoVision XT, Noldus, Wageningen, Netherlands).

The session was divided into three 5 min windows with photostimulation occurring throughout the middle 5 min window (473 nm light, 4 mW, 5 ms pulses at 20 Hz). The same laser and shutter setup described above was used except that a Master-8 pulse stimulator (A.M.P.I., Jerusalem, Israel) interfaced with the EthoVision XT system was used to drive laser pulses. Acetic acid (0.03%) was used to wipe and clean the chamber between animals.

We analyzed data using linear mixed effects models. Fixed effects included virus (ChR2 vs. eYFP), laser status (On or Off), and time by 5 min window, as well as the interaction between virus and laser status. Random effects were modeled using random intercepts for each subject.

## Real-time place preference

To assess whether cholinergic stimulation was inherently reinforcing or pleasurable to the animals, mice underwent a real-time place preference (RTPP) test (*Matthews et al., 2016*). We attached fiber optic

patch cables to the implanted ferrules on mice. Mice were then placed in the center of a transparent Plexiglas chamber (50 × 53 cm) divided into left and right compartments using center dividers, with an open gap in the middle allowing mice to freely access both compartments. The chamber was illuminated with 30 lux ambient light. Mice were allowed to freely move between compartments for 45 min during which entry into one of the two sides resulted in continuous photostimulation (473 nm light, 4 mW, 5 ms pulses at 20 Hz). The side paired with photostimulation was counterbalanced between animals and activity was averaged across 2 d. A video camera was placed directly above the arena to track mouse movement (EthoVision XT) and trigger photostimulation. Acetic acid (0.03%) was used to wipe and clean the chamber between animals. The percent of time mice spent on the side on which the laser was on was averaged over 2 d and compared between ChR2 and eYFP mice using a *t*-test.

## GCaMP photometry of basal forebrain cholinergic neurons

### Stereotaxic surgery for GCaMP photometry from basal forebrain cholinergic neurons

To measure neural activity from basal forebrain cholinergic neurons, we injected 1000 nL of an adeno-associated virus encoding the genetically encoded calcium sensor GCaMP (AAVdj/EF1a-DIO-GCaMP6s; Stanford Vector Core) into the basal forebrain of ChAT::Cre mice (AP –0.7; ML ± 1.75; DV –5.1 and –4.3, 500 nL at each depth). An optic fiber was implanted over the basal forebrain (AP –0.7; ML 1.75; DV –4.5).

### GCaMP photometry setup

The hardware setup for acquisition of bulk calcium fluorescence from multiple sites was adapted from *Kim et al., 2016*. The setup allowed for excitation of the sample at two wavelengths (405 and 470 nm) and collection of fluorescence emission at 525 nm. The excitation path consisted of a 405 nm and a 470 nm LED (Thorlabs M405FP1 and M470F3) which were collimated (Thorlabs F671SMA-405) and coupled to 400 nm and 469 nm excitation filters (Thorlabs FB400-10 and MF469-35), respectively. Compared to 470 nm excitation, 405 nm excitation of GCaMP is closer to the isobestic wavelength for calcium-dependent and calcium-independent GCaMP fluorescence, and thus was used to assess movement- and autofluorescence-related noise. Light from these two excitations sources was combined into one path via dichroic mirrors and filled the back aperture of a ×20 air objective (Nikon CFI Plan Apo Lambda). A fiber optic patch cord (Doric) containing optic fibers bundled into a single ferrule (400 μM diameter, 0.48 NA for each fiber) was positioned at the working distance of the objective. The end of the patch cable was connected to implanted ferrules on the animals' head. Emission resulting from the 405 or 470 nm excitation was split by a dichroic mirror, passed through a 525 nm emission filter (Thorlabs MF525- 39), and focused through a tube lens (Thorlabs AC254-100-A-ML) onto the face of CMOS camera (Hamamatsu ORCA-Flash V2). Frames were captured at 40 Hz and each LED was modulated at 20 Hz in an alternating fashion, resulting in a 20 Hz sample rate in the reference (405 nm excitation) and signal (470 nm excitation) channels. LED and camera timing as well as recording of timestamps from behavioral equipment was achieved using a data acquisition board (National Instruments NI BNC-2110). The system was controlled through custom MATLAB scripts modified from those made available by *Kim et al., 2016*. Prior to the start of each session, the entire system was shielded from outside light using blackout cloth.

### Behavior during GCaMP photometry

To investigate whether basal forebrain cholinergic neural activity increased during conditioning responding, mice first underwent uncued head-fixed training. Mice were rewarded for licking during unsignaled 3 s *Windows of Opportunity*. Rewards were delivered after a 0.5 s delay from the first lick in a *Window of Opportunity* to account for the slow dynamics of GCaMP6s. Mice were not punished for licking during the unrewarded ITIs.

Following recordings during the uncued stage of training, mice were then transitioned to the Cued Tone task described above. Mice were now rewarded for licking following the onset of tone cues, and rewards were delivered after a 0.5 s delay from the first lick within 3 s of tone onset.

## Photometry analysis

Photometry fluorescence traces were filtered using a 60 s median filter to extract an estimate of baseline fluorescence by accounting for bleaching and low-frequency fluctuations. The residual trace was filtered with a third-order median filter to eliminate single time point artifacts. DF/F was obtained by taking the difference between the residual trace and the baseline estimate and dividing by the baseline estimate, which was then expressed in percent. Fluorescence levels were compared as the area under the curve in a time window before (−2 to −1.5 s) and after (0–0.5 s) events. Fixed effects for linear mixed effects model analysis included wavelength (470 nm signal vs. 405 nm reference), time point (before or after event), and event type, and first- and second-order interactions. Random effects were modeled using random intercepts for each subject. Post hoc tests corrected for multiple comparisons within each session.

## ACh sensor measurements

### Stereotaxic surgery for ACh sensor measurements

To measure local levels of ACh, we injected 400 nL of an adeno-associated virus encoding the genetically encoded ACh sensor GACh3.0 (AAV9/hSyn-Ach4.3, ViGene) (Jing et al., 2018) into the BLA (AP −1.4; ML ± 3.1; DV −5.2). In order to photostimulate basal forebrain cholinergic terminals projecting to the BLA, an anterograde virus driving expression of the red-shifted opsin ChrimsonR (AAV5/Syn-FLEX-rc[ChrimsonR-tdTomato]) was injected into the basal forebrain (500 nL, AP −0.4; ML ± 1.8; DV −4.7). For control mice, a virus encoding just the fluorophore (AAV1/CAG-FLEX-tdTomato) was injected with the same volume at the same coordinates. An optic fiber (400 um diameter) was then implanted over the BLA (AP −1.4; ML ± 3.25; DV −4.8) in order to provide optical access to both record GACh3.0-related fluorescence using 470 nm blue light as well as photostimulate basal cholinergic forebrain cholinergic terminals in the BLA using interleaved 589 nm yellow light. Data was analyzed similar to the GCaMP photometry experiments, except that there was only fluorescence data from one channel (470 nm).

### Behavior during ACh sensor measurements

To investigate whether local ACh levels changed in the BLA during conditioning responding, mice first underwent uncued head-fixed training. Mice were rewarded for licking during unsignaled 3 s *Windows of Opportunity*. Rewards were delivered after a brief (0.1 s) delay from the first lick in a *Window of Opportunity*. Mice were not punished for licking during the unrewarded ITIs.

Similar to optogenetic experiments, mice were then tested during a Photostim-Unreinforced session and two Photostim-Reinforced sessions. Results are presented from the second Photostim-Reinforced session. During a subsequent session, we again varied photostimulation parameters by randomly delivering photostimulation on *Windows of Opportunity* using either 1, 2, or 10 5 ms pulses at 20 Hz, corresponding to up to 0.5 s of photostimulation. These pulse trains were also delivered at the onset of 3 s *Windows of Opportunity*.

Mice were then trained for three sessions on the cued tone task described above. Mice were rewarded for licking following the onset of tone cues, and rewards were delivered after a brief (0.1 s) delay from the first lick within 3 s from tone onset.

### Concurrent optogenetic photostimulation and measurement of local ACh levels using a genetically encoded fluorescent sensor

The hardware setup for acquisition of bulk ACh fluorescence from multiple sites was similar to that for GCaMP photometry. For concurrent optogenetic manipulation experiments, the system was modified to allow for 589 nm yellow light excitation through the same patch cable. An additional dichroic mirror combined the LED light paths with that of a 589 nm laser. The laser was powered on throughout the experiments to minimize intensity fluctuations and was modulated by opening/closing a mechanical laser shutter head (SR475, Stanford Research Systems), controlled by a shutter driver (SR474, Stanford Research Systems). 470 nm LED light was used to excite the ACh sensor, and 525 nm emitted photons were collected, for 25 ms at 20 Hz (every 50ms) using the filters described above. When photostimulation was provided using 589 nm laser light, it was delivered as 5 ms pulses at 20 Hz (4 mW) in between the 470 nm light pulses in order to prevent spectral cross-talk.

Fluorescence levels were analyzed as the area under the curve after photostimulation onset (0–0.5 s for reward, 1, 2, or 10 pulses; 0–2 s for 2 s photostimulation). Fixed effects for linear mixed effects model analysis for sessions with different durations of photostimulation included virus (ChrimsonR vs. tdTomato control fluorophore), photostimulation pulses (Reward, 1, 2, or 10 pulses), and their interaction. Random effects were modeled using random intercepts for each subject. Post hoc tests corrected for multiple comparisons included comparisons between ChrimsonR and tdTomato mice at each number of pulses, and comparisons within each virus group for 1, 2, or 10 pulses to Reward. For comparisons of evoked ACh in Photostim-Unreinforced and Photostim-Reinforced sessions, linear mixed effects model fixed effects included virus (ChrimsonR vs. tdTomato), session type (unreinforced or reinforced), and their interaction. Random effects were modeled using random intercepts for each subject.

## In vivo electrophysiology

### In vivo electrophysiology surgery

Mice were anesthetized using isoflurane gas (1–4%) and mounted on a stereotaxic apparatus (Kopf Instruments) to implant optrodes (i.e., combination of electrode and optical fiber). A midline incision was made down the scalp and craniotomies were opened using a dental drill. Optrodes were chronically implanted in either hemisphere in the basolateral amygdala (BLA) and the prelimbic (PL) subregion of the dorsomedial prefrontal cortex (dmPFC). The stereotaxic coordinates to target the BLA were –1.50 mm anterior–posterior (AP), ± 3.15 mm medial–lateral (ML), and –5.00 mm dorsal–ventral (DV). The stereotaxic coordinates to target PL were +1.80 AP, ± 0.35 mm ML, and –2.00 mm DV. In addition to these optrodes, optical fibers were implanted bilaterally in the basal forebrain (BF) to photostimulate ChR2-expressing ChAT neurons while recording neural activity from the BLA and dmPFC. The stereotaxic coordinates for the BF fibers were –0.40 mm AP, ± 1.80 mm ML, and –4.50 mm DV. All stereotaxic coordinates were calculated relative to Bregma. Finally, an aluminum bar was horizontally positioned behind lambda to provide anchoring points during head-fixed recordings. All these implants were secured to the skull using stainless steel self-drilling screws (Small Parts), adhesive cement (C&B Metabond, Parkell), and dental acrylic (Ortho-Jet, Lang Dental). At the end of the surgeries, incisions were sutured and postoperative analgesia and fluids were provided as needed. Mice were allowed to recover from surgery for at least 1 wk.

### In vivo electrophysiology recordings

Extracellular recordings were performed using in-house-built multichannel electrodes, each containing a 16-channel Omnetics connector, an optical fiber attached to the connector, and a low-resistance silver wire to provide ground. The microwire used for the electrodes was a 22.9 µm HML-insulated nichrome wire (Stablohm 675, California Fine Wire). Microwires were secured to the connector pins using a silver print coating (GC Electronics). All connections were then secured using dental acrylic. Serrated fine scissors were used to cut the tip of the microwires to a length of 500–1000 µm from the tip of the optical fiber. The microwire tips were then gold-plated to reduce impedance and improve signal-to-noise ratio (*Ferguson et al., 2009*). Gold plating was achieved by submerging the electrode tips in a solution containing equal parts of a non-cyanide gold solution (SIFCO Selective Plating) and a 1 mg/mL polyethylene glycol solution. A cathodal current of 1 µA was applied to individual channels to reduce impedances to a range of 200–300 kΩ.

Multichannel extracellular recording setups (Open Ephys) were used to monitor neural activity while mice performed behavioral tasks. Extracellular signals were recorded at 30 kHz with band-pass filters set at 0.2 and 7600 Hz. These raw signals were then processed offline to extract single-unit activity.

### In vivo electrophysiology spike extraction

A common reference was calculated for each 16 channel array by calculating the median trace across all channels. This trace was then subtracted from each channel on that array. Data was then filtered from 300 to 7000 Hz using a fourth-degree Butterworth filter applied using the filtfilt function in MATLAB. Spikes were identified by negative deviations greater than 7.4 times the median absolute deviation. Spikes were aligned to their minima and waveforms from 375 µs before the trough and 1000 µs after the trough were extracted for spike sorting. Single-unit waveforms were then sorted

using commercial software (Offline Sorter, Plexon Inc) by combining principal component and peak-trough voltage features in three-dimensional space. Neural firing properties were visually inspected, including autocorrelations and cross-correlations, in order to exclude multiunit activity or repeated recordings of the same unit across multiple wires in a bundle. Neurons were additionally inspected to ensure they did not represent behaviorally locked artifacts. Only neurons that fired at least 0.01 Hz across the whole session were included in analyses.

### In vivo electrophysiology analysis

Peri-event time histograms were constructed by extracting spiking activity around photostimulation in 10 ms bins from 8 s before to 16 s after photostimulation onset. Baseline activity was defined as the activity from 4 s before up to the onset of photostimulation. Activity for each trial was transformed using the mean and standard deviation of baseline activity across trials. Data was smoothed with a 50 ms Gaussian kernel. Activity within each area for each bin was compared using rank-sum tests to compare Photostim-Unreinforced and Photostim-Reinforced sessions, separately for ChR2 and eYFP mice.

Modulation around the time of photostimulation was calculated by comparing firing rates before photostimulation onset (−1 to 0 s) with firing rates shortly after photostimulation onset (0–0.5 s). Facilitation was classified as neurons with statistically significant increases in firing rates after photostimulation onset, and suppression was defined as neurons with statistically significant decreases in firing rates after photostimulation onset (sign-rank test, $p<0.01$). Proportions were compared using two-sided chi-square proportion tests, with p-values corrected for multiple comparisons using Holm's procedure.

### In vivo electrophysiology histology

Mice were anesthetized using isoflurane to mark the position of electrode tips by producing microlesions (anodal current, 25–40 µA for at least 20 s). Mice were then euthanized with sodium pentobarbital (150 mg/kg), and transcardially perfused with saline solution and 4% paraformaldehyde (PFA, pH 7.3). Brains were extracted and fixed in 4% PFA for at least 24 hr and then equilibrated in a 30% sucrose solution for 48 hr. Coronal sections were cut at 40 µm using a microtome (HM430, Thermo Fisher Scientific). Brain sections containing the BLA, dmPFC, and BF were mounted on microscope slides, and stained with 4',6-diamidino-2-phenylindole (DAPI). Images of the BLA, dmPFC, and BF were acquired using a confocal laser-scanning microscope (Olympus FV1000). Expression of eYFP and/or ChR2 was examined from the BF sections, whereas the location of microlesions was examined from the BLA and dmPFC sections to determine the neural recording sites. These were reconstructed onto coronal drawings adapted from a mouse brain atlas (*Franklin, 2008*).

## Ex vivo electrophysiology

### Ex vivo electrophysiology surgery

We generated double transgenic mice by crossing homozygous ChAT::Cre mice with hemizygous VGAT::flpo mice. Double hemizygous offspring (ChAT::Cre × VGAT::flpo mice), confirmed by genotyping (Transnetyx) were used for these experiments. In a first surgery, for optogenetic photostimulation of basal forebrain cholinergic neurons, we injected 500 nL of an adeno-associated virus encoding channelrhodopsin (AAV5/EF1a-DIO-hChR2(H134R)-mCherry; UNC, $3.5 \times 10^{12}$) bilaterally into basal forebrain of each hemisphere at AP –0.4, ML ± 1.8, DV –5. For fluorescent labeling of BLA GABAergic neurons, we injected 400 nL of an adeno-associated virus encoding the fluorophore eYFP (AAV5/EF1a-fDIO-eYFP-WPRE; UNC, $3.8 \times 10^{12}$) bilaterally into the BLA at AP –1.6, ML ± 3.3, DV –5.1. After at least 6 wk, we performed a second surgery to label BLA neurons projecting to the medial prefrontal cortex (mPFC) through injection of a retrograde tracer (200 nL cholera toxin b subunit fused to Alexa-647, CTB-647) in the mPFC (AP 1.8, ML ± 0.35, DV –2).

### Ex vivo electrophysiology recordings

One week after the CTB-647 surgery, mice were deeply anaesthetized via I.P. injection of sodium pentobarbital (200 mg/kg) prior to transcardial perfusion with 20 mL ice-cold modified artificial cerebrospinal fluid (ACSF; composition in mM: 75 sucrose, 87 NaCl, 25 NaHCO$_3$, 2.5 KCl, 1.3 NaH$_2$PO$_4$

* H₂O, 7 MgCl₂ * 6H₂O, 0.5 CaCl₂ * 2H₂O, 5 ascorbic acid, in ddH₂O; osmolarity 324–328 mOsm, pH 7.3–7.4) saturated with carbogen gas (95% oxygen, 5% carbon dioxide). The brain was quickly dissected out of the cranial cavity and a semiautomatic vibrating blade microtome (VT1200; Leica, IL) was then used to prepare 300-µm-thick coronal slices containing the BLA. Brain slices encompassing the basal forebrain and mPFC were also collected for verification of ChR2-mCherry and CTB-647 expression, respectively. Slices were then transferred to ACSF (composition in mM: 126 NaCl, 2.5 KCl, 26 NaHCO₃, 1.25 NaH₂PO₄ * H₂O, 1 MgCl₂ * 6H₂O, 2.4 CaCl₂ * 2H₂O, and 10 glucose, in ddH₂O; osmolarity 299–301 mOsm; pH 7.30–7.40) saturated with carbogen gas (95% oxygen, 5% carbon dioxide) in a water bath kept at 30–32°C. They were allowed to recover for at least 1 hr before transfer to the recording chamber for electrophysiological recordings.

In the recording chamber, BLA slices were continually perfused with carbogen-saturated ACSF, at a temperature of 31 ± 1 °C, via a peristaltic pump (Minipuls3; Gilson, WI). BLA neurons were visualized through an upright microscope (Scientifica, UK) using infrared differential interference contrast (IR-DIC) optics and a Q-imaging Retiga Exi camera (Q Imaging, Canada). Identification of eYFP+ GABAergic neurons and CTB-647-expressing mPFC-projector neurons was achieved through a 40× water-immersion objective using brief fluorescence illumination from a 470 nm LED light source (pE-100; CoolLED, NY) or a metal halide lamp (Lumen 200, Prior Scientific Inc, MA), respectively, through appropriate excitation/emission filters (Olympus, PA). Thin-walled borosilicate glass capillary tubing was shaped into microelectrodes for recording using a horizontal puller (P-97, Sutter Instruments, CA) and had resistance values of 3–6 MΩ when filled with internal solution (composition in mM: 125 potassium gluconate, 20 HEPES, 10 NaCl, 3 MgATP, and 0.1% neurobiotin; pH 7.30–7.33; 286–287 mOsm). Recorded signals were amplified with a Multiclamp 700B amplifier (Molecular Devices, CA), low-pass filtered at 3 kHz, digitized at 10 kHz using a Digidata 1550, and recorded using pClamp 10.4 software (Molecular Devices). Series resistance, input resistance, and holding current were monitored throughout experiments via a 5 mV, 0.1 s step. Any significant changes were interpreted as signs of cell deterioration and recordings were terminated. To assess the response of BLA neurons to cholinergic input, ChR2-expressing cholinergic terminals were activated by trains of 5 ms pulses of 470 nm light (pE-100; CoolLED) delivered through the 40× objective every 30 s, while recording in current-clamp mode. To isolate monosynaptic currents tetrodotoxin (TTX; 1 µM; Tocris, MN) and 4-aminopyridine (4AP; 1 mM, Sigma, MO) were included in the ACSF (*Petreanu et al., 2007*). Scopolamine (10 µM; Sigma) was used to block muscarinic ACh receptors, and a cocktail of nicotinic antagonists dihydro-ß-erythroidine (10 µM, Tocris), methyllycaconitine (0.1 µM; Sigma), and mecamylamine (10 µM; Sigma) was used to block nicotinic ACh receptors. Offline analysis of peak current amplitude was performed in Clampfit 10.4 (Molecular Devices). Capacitance and membrane resistance (Rm) were calculated from a 5 mV, 0.1 s hyperpolarizing step in voltage clamp using custom MATLAB software written by Praneeth Namburi based on MATLAB implementation of the Q-method (*Novák and Zahradník, 2006*).

### Ex vivo electrophysiology histology

Following recording, slices were fixed in 4% PFA overnight at 4°C then washed in PBS (4 × 10 min). To reveal neurobiotin-labeled cells, recorded slices were blocked in PBS containing 0.3% Triton (PBS-T 0.3%) and 3% normal donkey serum (NDS; Jackson ImmunoResearch, USA) at room temperature for 60 min before incubation in PBS-T 0.3% with 3% NDS and CF405-conjugated streptavidin (1:1000; Biotium, CA). After 90 min, slices were washed in PBS (4× 10 min) then mounted onto glass slides and coverslipped using polyvinyl alcohol mounting medium with DABCO (Sigma). Images of the BLA were captured using a confocal laser scanning microscope (Olympus FV1000), with FluoView software (Olympus). Neurobiotin-filled recorded neurons were imaged at high magnification through a 40×/1.3 NA oil-immersion objective using serial z-stacks with an optical slice thickness of 3 µm.

### Immunohistochemistry

After behavioral experiments, mice were anaesthetized with pentobarbital sodium (200 mg/kg) and transcardially perfused with ice-cold Ringer's solution followed by ice-cold 4% PFA in PBS (pH 7.3). Brains were extracted and fixed in 4% PFA for 24 hr and equilibrated in 30% sucrose in PBS for 3 d. Coronal sections were cut at 40 µm using a sliding microtome (HM430; Thermo Fisher Scientific) and kept in PBS at 4°C until antibody staining.

Sections from the PFC, BF, and BLA were taken for immunohistochemistry. Sections were washed six times for 5 min each in 1× PBS, and then blocked for 2 hr at room temperature in 3% normal donkey serum in 0.3% PBS-Triton. Sections were then incubated overnight at room temperature in blocking solution with primary antibody goat-anti-ChAT (1:200/1:100). The next day, sections were washed six times for 5 min in 1× PBS. Sections were then incubated in secondary antibody donkey-anti-goat 647 (1:500) in blocking solution at room temperature for 2 hr. Finally, sections were washed six times for 5 min in 1× PBS and mounted on microscope slides using a Vectashield mounting medium with DAPI. Slides were sealed with clear nail polish to preserve sections. All washing and staining steps were done covered on a shaker.

Confocal microscopy fluorescent images were captured using a confocal laser scanning microscope (Olympus FV1000), with FluoView software (Olympus), under a 10×/0.40 NA dry objective or a 40×/1.30 NA oil immersion objective. Images were subsequently processed in Adobe Illustrator (Adobe Systems Incorporated, CA). Similar to prior work, we also observed some ectopic expression of ChAT (*Hedrick et al., 2016*), particularly in later litters of breeding pairs, and excluded these mice based on postmortem histological examination.

## Acknowledgements

We thank C Wildes, R Wichmann, J Wang, A Heynen, and L Keyes for technical assistance. We thank V Breton-Provencher, J Shih, M Sur, and the members of the Bear lab for additional support. We thank A Sutton, M Picciotto, R Crouse, A Kepecs, F Sturgill, and E Calipari for helpful discussions. EYK was supported by NIH grant K08-MH116135 (NIMH). CAS was supported by an NIH grant R00 DA04510 (NIDA), a NARSAD Young Investigator Award (Brain and Behavior Research Foundation), and by an Alkermes Pathways Research Award grant, an independent competitive grants program supported by Alkermes. MF was supported by NIH grant K99-EY029326 (National Eye Institute). DBP was supported by R01-DC017078 and funding from the Nancy Lurie Marks Family Foundation. KMT is an HHMI investigator and Wylie Vale Professor at the Salk Institute and was supported by funding from the JPB Foundation, New York Stem Cell Foundation, HHMI, Kavli Foundation, R01-MH102441 (NIMH), the NIH Director's New Innovator Award DP2-DK102256 (NIDDK), Pioneer Award DP1-AT009925 (NCCIH), and R37-MH102441 (NIMH).

## Additional information

### Funding

| Funder | Grant reference number | Author |
|---|---|---|
| NIH Office of the Director | DP2-DK102256 | Kay M Tye |
| NIH Office of the Director | K08-MH116135 | Eyal Y Kimchi |
| NIH Office of the Director | DP1-AT009925 | Kay M Tye |
| NIH Office of the Director | R01-MH102441 | Kay M Tye |
| NIH Office of the Director | R00 DA04510 | Cody Siciliano |
| NIH Office of the Director | R00-EY029326 | Ming-fai Fong |
| NIH Office of the Director | R01-DC017078 | Daniel B Polley |
| Howard Hughes Medical Institute | | Kay M Tye |
| Nancy Lurie Marks Family Foundation | | Daniel B Polley |
| JPB Foundation | | Kay M Tye |
| New York Stem Cell Foundation | | Kay M Tye |

| Funder | Grant reference number | Author |
|---|---|---|
| Kavli Institute for Brain and Mind, University of California, San Diego | | Kay M Tye |
| NIH Office of the Director | R37-MH102441 | Kay M Tye |

The funders had no role in study design, data collection and interpretation, or the decision to submit the work for publication.

## Author contributions

Eyal Y Kimchi, Conceptualization, Data curation, Software, Formal analysis, Supervision, Funding acquisition, Validation, Investigation, Visualization, Methodology, Writing – original draft, Project administration, Writing – review and editing; Anthony Burgos-Robles, Data curation, Investigation, Writing – original draft, Writing – review and editing; Gillian A Matthews, Investigation, Methodology, Writing – original draft; Tatenda Chakoma, Makenzie Patarino, Validation, Investigation, Writing – original draft; Javier C Weddington, Wannan Yang, Shaun Foutch, Renee Simons, Ming-fai Fong, Miao Jing, Investigation; Cody Siciliano, Methodology, Writing – original draft; Yulong Li, Resources, Methodology; Daniel B Polley, Funding acquisition, Writing – review and editing; Kay M Tye, Conceptualization, Supervision, Funding acquisition, Methodology, Writing – original draft, Project administration, Writing – review and editing

## Author ORCIDs

Eyal Y Kimchi ⓘ https://orcid.org/0000-0003-4327-1102
Anthony Burgos-Robles ⓘ https://orcid.org/0000-0002-6729-2648
Gillian A Matthews ⓘ https://orcid.org/0000-0001-6754-0333
Cody Siciliano ⓘ https://orcid.org/0000-0001-9871-2089
Ming-fai Fong ⓘ https://orcid.org/0000-0002-2336-4531
Daniel B Polley ⓘ https://orcid.org/0000-0002-5120-2409
Kay M Tye ⓘ https://orcid.org/0000-0002-2435-0182

## Ethics

All experiments involving the use of animals were in accordance with National Institutes of Health guidelines and approved by the Massachusetts Institute of Technology's Committee on Animal Care, protocol number 0118-002-21.

Reviewer #1 (Public Review): https://doi.org/10.7554/eLife.89093.2.sa1
Reviewer #2 (Public Review): https://doi.org/10.7554/eLife.89093.2.sa2
Reviewer #3 (Public Review): https://doi.org/10.7554/eLife.89093.2.sa3

# Additional files

## Supplementary files
• MDAR checklist

## Data availability

Custom MATLAB code for controlling GCamp Photometry Set up are publicly available under a GNU General Public License v.3.0 on GitHub (copy archived at *Tyelab, 2023*). Source data needed to recreate the primary statistical results shown in Figures and Supplementary Figures have been provided as source data files. Electrophysiological recordings are available in Dryad.

The following dataset was generated:

| Author(s) | Year | Dataset title | Dataset URL | Database and Identifier |
|---|---|---|---|---|
| Kimchi EY, Burgos-Robles A, Matthews GA, Chakoma T, Patarino M, Weddington JC, Siciliano C, Yang W, Foutch S, Simons R, Fong M, Jing M, Li Y, Polley DB, Tye KM | 2024 | Reward contingency gates selective cholinergic suppression of amygdala neurons | https://doi.org/10.5061/dryad.dbrv15f85 | Dryad Digital Repository, 10.5061/dryad.dbrv15f85 |

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
