## [Editor Report · eLife assessment]

This **valuable** article examines the role of basal forebrain cholinergic (ACh) projection neurons and their inputs to the basolateral amygdala (BLA) and effects on BLA activity during reward seeking. The article provides **compelling** evidence that ACh may have different effects on network activity in the BLA depending on the state of the network during reward engagement, whereas behavioral data indicating that these ACh neurons/inputs are involved in uncued reward seeking specifically is somewhat less complete. The article will be of interest to those studying amygdala circuitry, reward processing, and neuromodulation broadly defined.

---

## [Referee Report · Reviewer #1 (Public Review)]

In their manuscript entitled, "Reward contingency gates selective cholinergic suppression of amygdala neurons," Kimchi and colleagues explore the engagement and consequences of acetylcholine (ACh) signaling in the basolateral amygdala (BLA) using a number of sophisticated methodological approaches.

Perhaps the most compelling new idea in this manuscript is that ACh may have different effects on network activity in the BLA, a conclusion based on the measurement of equivalent photo-stimulated ACh levels in BLA during rewarded vs. unrewarded lick bouts despite increased licking/consumption in the rewarded bouts. The authors hypothesize that, "this could suggest that reward associations may gate post-synaptic responses to photostimulation." The electrophysiological data showing that overall firing of BLA neurons during licking was higher as a result of photostimulation during unreinforced, and lower as a result of photostimulation during reinforced, sessions is intriguing in this context, as is the contrast with the overall ACh-mediated stimulation of firing in dorsomedial prefrontal cortex. The ex-vivo data presented showing that ACh depresses BLA neuron activity via muscarinic ACh receptors on glutamate neurons and nicotinic ACh receptors on GABA neurons, along with previous data in the field suggesting that ACh has divergent effects on neuronal firing rate depending on whether baseline firing is low (tonic) or high (phasic), provides intriguing hints as to the role of ACh in state-dependent modulation of BLA activity.

One of the primary questions that came up while reading this manuscript was what behavioral domains were being measured with the "windows of opportunity" task. As noted by the authors, the cholinergic system has been implicated in arousal, reward thresholds, motivation and many other behaviors that might alter performance in this task, complicating interpretation of the data presented. In addition, some additional details of the task are needed for the field to be able to replicate these experiments.

---

## [Referee Report · Reviewer #2 (Public Review)]

Kimchi et al. examined the role of cholinergic inputs to the amygdala in regulating reward-seeking behavior. To investigate this, the authors developed a head-fixed behavioral task where animals were trained to lick at random intervals, with some of these responses being reinforced ("windows of opportunity") as opposed to control epochs when no reward was delivered.

The authors conducted in vivo optogenetic stimulation of basal forebrain cholinergic neurons and discovered that a 2-second optical stimulation of these neurons encouraged licking behavior when followed by reward delivery. This was in comparison to time epochs where no reward was delivered or compared to control mice only expressing EYFP. However, it remained unclear how many trials were required for this effect to manifest.

Furthermore, they demonstrated that the stimulation of basal forebrain cholinergic neurons did not induce real-time place preference or affect locomotion. The reward-driven licking behavior was also mitigated by systemic cholinergic receptor antagonists.

Next, the authors observed the bulk calcium dynamics from these neurons in a version of the task where an auditory cue predicted reward availability. They found strong calcium signals when mice were licking and when the tone was present, but also reported signals when mice were spontaneously licking.

By injecting a genetically encoded Acetylcholine (Ach) sensor directly into the Basolateral Amygdala (BLA), they showed that Ach signals were present when mice were engaged in licking, both during reward availability and for non-rewarded licks. Photostimulation of Ach terminals directly in the BLA increased licking behavior when a reward was available.

Finally, using in vivo and ex vivo physiology, they demonstrated that Ach signaling influences the electrophysiological dynamics in the BLA. This may help clarify some of the postsynaptic responses triggered by this neuromodulator.

Strengths of the paper:

1. The experiments were well-executed and sufficiently powered, with most statistics being correctly reported.

2. The paper is a technical tour de force, employing fiber photometry, in vivo and ex vivo electrophysiology, optogenetics, and behavioral approaches.

3. Robust effects were observed in most of the experiments.

Weaknesses:

1. The experimental design varies slightly across each behavioral experiment, making it difficult to directly compare one effect to another.

2. The paper doesn't include data showing the precise location for the Ach recordings. As a result, it is unclear whether these signals are specific to the BLA, or whether they might also be coming from neighboring regions.

---

## [Referee Report · Reviewer #3 (Public Review)]

This important manuscript investigates the role of basal forebrain cholinergic interneurons in conditioned responding by measuring the licking behaviour of head-fixed mice during photostimulation of the aforementioned neurons. Licking is found to increase only during windows when licking is rewarded, and similar behaviour is observed when terminals are stimulated in basolateral amygdala, then several more experiments are conducted to determine the behavioural and anatomical specificity of the effect. The findings are solid, particularly those relating to the recordings, although the interpretation of the behavioural findings is still somewhat unclear.

Strengths

• The manuscript is beautifully written and structured. I found it really easy to follow and felt that the authors did an exceptional job of walking me through each experiment that they completed, the rationale for it, and what they found.

• The question of the function of basal forebrain cholinergics is an interesting one and a somewhat understudied question, so the study is timely and on an interesting topic.

• The experiments are well-designed and the findings are novel. There are a number of important control experiments performed to determine that the observed effects were not due to locomotor activity and that stimulating basal forebrain ACh neurons is not inherently reinforcing.

• The discussion is really nice - covering important topics such as potential interactions with dopamine, the potential anatomical specificity of the effects observed, and the possibility that projections other than those studied here might mediate effects, among other things.

Weaknesses

• Although very clearly written and set out, I found myself confused by the behavioural findings and their interpretation. Mainly this was because photostimulation only increased licking during the window of opportunity, which is not signalled by any discrete stimulus, which means that the only signal that the animal receives to determine that they are within the reward window is them receiving the reward. Therefore, the only time within this window that licking could be increased is post-reward (otherwise the reward window is identical to a non-rewarded window) and it is not clear to me what this increase in post-award licking might mean? In fact, this time post-award is actually the time the animal is most certain to not receive another reward for a few seconds, meaning that licking at this time is not a useful behaviour and therefore it is difficult to interpret what it means to artificially increase licking at this time. I think it would probably have been less confusing for the authors to study a paradigm in which animals develop a conditioned response that is unsignaled by discrete stimuli and then to inhibit basal forebrain ACh prior to that response.

• I should also note that the authors state (Lines 249-251) that stimulation increases responding prior to reinforcer delivery, but I couldn't find evidence for this, and it seems counterintuitive to me that it would do so because then how would the animals discriminate the window of opportunity from a non-rewarded window? Perhaps I misunderstood something, but I found this confusing.

• I do not think the behaviour in this task can be classed as operant - it is still a good task and still fine for detecting conditioned responding, but it cannot determine whether the responding is governed by a response-outcome association in the absence of a stimulus-outcome association (with stimuli being the licking spout, other facets of the behavioural context etc) through bidirectionality or omission, as would be required to demonstrate its operant nature.

• I was confused by the pupil dilation data in Figure S4 as the authors seem to want to argue that this effect, although specific to the rewarded window as licking is, is independent of the licking behaviour as it develops more slowly than the behaviour (Lines 201-202). I was curious as to how the authors interpret these data then? Does it indicate that stimulating basal forebrain ACh interneurons does both things (i.e. increases arousal AND conditioned responding in the absence of discrete stimuli) but that the two things are independent of each other?

• The authors refer to the dorsal medial prefrontal cortex in mice, which from the methods appears to be the prelimbic region. My understanding is that dmPFC has fallen out of favour for use in mice as it is not homologous to the same region in primates and can be confusing for this reason.